# Essential protein P116 extracts cholesterol and other indispensable lipids for *Mycoplasmas*

**Lasse Sprankel[1,7], David Vizarraga [2,7], Jesús Martín[2], Sina Manger[1], Jakob Meier-Credo [3], Marina Marcos[4], Josep Julve [5], Noemi Rotllan[5], Margot P. Scheffer[1], Joan Carles Escolà-Gil [5], Julian D. Langer [3,6], Jaume Piñol[4], Ignacio Fita [2] ✉ & Achilleas S. Frangakis [1] ✉**

*Mycoplasma pneumoniae*, responsible for approximately 30% of community-acquired human pneumonia, needs to extract lipids from the host environment for survival and proliferation. Here, we report a comprehensive structural and functional analysis of the previously uncharacterized protein P116 (MPN_213). Single-particle cryo-electron microscopy of P116 reveals a homodimer presenting a previously unseen fold, forming a huge hydrophobic cavity, which is fully accessible to solvent. Lipidomics analysis shows that P116 specifically extracts lipids such as phosphatidylcholine, sphingomyelin and cholesterol. Structures of different conformational states reveal the mechanism by which lipids are extracted. This finding immediately suggests a way to control *Mycoplasma* infection by interfering with lipid uptake.

*Mycoplasma pneumoniae* is a facultative intracellular human pathogen that causes community-acquired pneumonia that can result in severe systemic effects[1]. Unlike other respiratory pathogens, there is no approved vaccine against *M. pneumoniae*[2]. *Mycoplasma* species lack a cell wall and have the smallest known genomes[3]. *M. pneumoniae*, with a 816-kb genome, is a model organism for a minimal cell[4]. Many of the metabolic pathways that are required to synthesize essential products are absent, which makes uptake by specialized mechanisms necessary. In fact, *M. pneumoniae* cannot synthesize several of the lipids that are important components of the cell membrane, such as sphingomyelin, phosphatidylcholine and cholesterol[5]. Instead, it must take up lipids from the host environment, and it adapts its membrane composition depending on the medium in vitro[6–8]. Cholesterol in particular, which is present in only a few prokaryotes, is essential for *M. pneumoniae* cells and several other *Mycoplasma* species[6]. It is the most abundant

lipid in the membranes, accounting for 35–50% of the total lipid fraction[6]. Comprehensive studies on other cholesterol-utilizing bacteria are largely lacking; the best characterized organism in this group is *Mycobacterium tuberculosis*, for which it has been proposed that an ABC transporter homolog and other genes from the *mce4* operon are involved in cholesterol uptake[9]. *M. tuberculosis* uses cholesterol as a carbon source, enabling long-term infections with the bacteria[10]. It has been shown that *M. pneumoniae* survive long-term in cholesterol-rich atherosclerotic plaques[11]. For other clinically relevant bacteria that use cholesterol, like *Borrelia burgdorferi* or *Helicobacter pylori*, the uptake mechanism remains elusive[12]. To date, it is unclear how *Mycoplasma* spp. uptake lipids from the environment.

In this work, we report the structural and functional characterization of P116. This protein was originally reported to contribute to host-cell adhesion. Furthermore, P116 is an essential protein for the

[1]Buchmann Institute for Molecular Life Sciences and Institute of Biophysics, Goethe University Frankfurt, Frankfurt, Germany. [2]Instituto de Biología Molecular de Barcelona (IBMB-CSIC), Parc Científic de Barcelona, Barcelona, Spain. [3]Proteomics, Max Planck Institute of Biophysics, Frankfurt, Germany. [4]Institut de Biotecnologia i Biomedicina and Departament de Bioquímica i Biologia Molecular, Universitat Autònoma de Barcelona, Cerdanyola del Vallès, Spain. [5]Institut de Recerca de l'Hospital de la Santa Creu i Sant Pau and CIBER de Diabetes y Enfermedades Metabólicas Asociadas (CIBERDEM), Barcelona, Spain. [6]Proteomics, Max Planck Institute for Brain Research, Frankfurt, Germany. [7]These authors contributed equally: Lasse Sprankel, David Vizarraga. ✉e-mail: ifrcri@ibmb.csic.es; achilleas.frangakis@biophysik.org

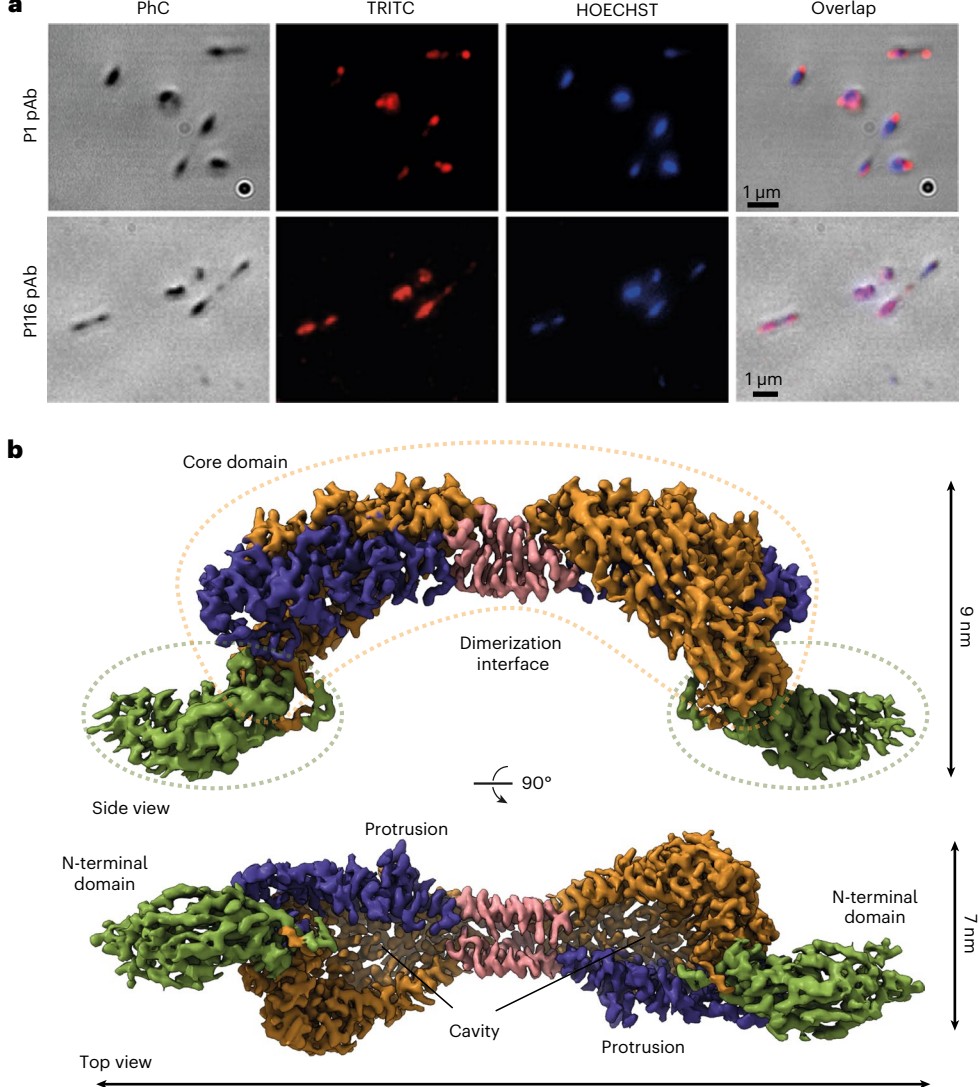

**Fig. 1 | Localization of P116 in *Mycoplasma pneumoniae* cells and structure of P116. a**, Phase contrast (PhC) immunofluorescence microscopy images *of M. pneumoniae* cells using labeling with polyclonal antibodies (pAb) against the ectodomains of adhesin P1 (top row; used as a reference) and P116 (bottom row) (two separate experiments with independent samples were performed). Labeling for P1 concentrates at the tip of the cell; for P116, it covers the whole surface homogenously. **b**, Two views of the cryoEM density map of the complete extracellular region of the P116 dimer at 3.3-Å resolution (from 1.3 million particles), 90° apart. The homodimer is held together by the dimerization interface (shown in pink). The core domains have four contiguous antiparallel helices (shown in blue) and a β-sheet with five antiparallel strands (shown in orange). The N-terminal domain is shown in green. The top view displays a huge cavity that is fully accessible to solvent. The cleft providing access to the cavity spans the whole core domain. Each monomer also has a distinct protrusion (shown in blue as part of the antiparallel α-helices).

viability of *M. pneumoniae* cells and is strongly immunogenic, thus making it a promising target for therapeutics[13]. Despite the essential role of P116, the *M. pneumoniae* genome contains only a single copy of *mpn_213* (gene encoding P116), and, on average, only 34 copies of the protein are present in *M. pneumoniae*[14]. By contrast, the most immunogenic protein, P1, is not essential, has multiple gene copies present in the genome[15], and has a 20-fold-higher copy number[14]. To elucidate the role of P116, we first determined the structure of the ectodomain by single-particle cryo-electron microscopy (cryoEM). To the best of our knowledge, this structure represents a previously uncharacterized fold (with no matches in the Protein Data Bank) featuring a uniquely large hydrophobic cavity that is fully accessible to solvent. Using mass spectrometry, we identified several different lipids (including cholesterol) bound to P116, some of which are essential, matching observed densities in the hydrophobic cavity. On the basis of these findings,

we describe the mechanism by which *Mycoplasma* spp. extract lipids from the environment and possibly also deposit them in their own membrane, thus explaining the essential role of P116 in the survival of *M. pneumoniae* cells.

## Results

### P116 is evenly distributed on the cell surface

A construct predicted to span the whole ectodomain of P116 from *M. pneumoniae* (residues 30–957) was overexpressed in *Escherichia coli* and purified by His-tag affinity and gel filtration chromatography (Methods and Extended Data Fig. 1). Immunolabeling with both polyclonal and monoclonal antibodies against this construct showed an intense and uniform distribution of labeling across the whole surface of the *M. pneumoniae* cells (Fig. 1a), and adhesion and motility were unaffected by the antibodies (Supplementary Table 1 and Supplementary

**Table 1 | CryoEM data collection, refinement and validation statistics**

| | P116<br>(EMD-15274) | P116 empty<br>(EMD-15275) | P116 refilled<br>(EMD-15276) | P116+HDL<br>(EMD-15277) |
|---|---|---|---|---|
| **Data collection and processing** | | | | |
| Microscope | FEI Titan Krios | FEI Titan Krios | FEI Titan Krios | FEI Titan Krios |
| Detector | Gatan K2 Summit | Gatan K3 Summit | Gatan K2 Summit | Gatan K2 Summit |
| Acquisition Software | SerialEM 3.8 | EPU 2.12 | SerialEM 3.8 | SerialEM 3.8 |
| Magnification | ×130,000 | ×105,000 | ×130,000 | ×130,000 |
| Voltage (kV) | 300 | 300 | 300 | 300 |
| Electron exposure ($e^-/Å^2$) | 50 | 50 | 50 | 50 |
| Defocus range (μm) | −1 to −3.5 | −1 to −3.5 | −1 to −3.5 | −1 to −3.5 |
| Pixel size (Å) | 1.05 | 0.831 | 1.05 | 1.05 |
| Symmetry imposed | $C_1$ | $C_1$ | $C_1$ | $C_1$ |
| Initial particle images | 3,463,490 | 4,532,601 | 2,930,863 | 262,981 |
| Final particle images | 1,315,362 | 633,322 | 1,311,526 | 46,277 |
| Map Resolution (Å) | 3.3 | 4 | 3.5 | 8.6 |
| FSC threshold | 0.143 | 0.143 | 0.143 | 0.143 |
| Map resolution range (Å) | 2.5–5 | 3.5–6 | 3–7 | 7–12 |
| Number of frames | 34 | 50 | 29 | 30 |
| Micrographs used | 4,367 | 15,299 | 4,019 | 3,114 |
| Processing software | cryoSPARC v3.3.2 | cryoSPARC v3.3.2 | cryoSPARC v3.3.2 | cryoSPARC v3.3.2 |
| Motion correction | cryoSPARC v3.3.2 | cryoSPARC v3.3.2 | cryoSPARC v3.3.2 | cryoSPARC v3.3.2 |
| CTF estimation | cryoSPARC v3.3.2 | cryoSPARC v3.3.2 | cryoSPARC v3.3.2 | cryoSPARC v3.3.2 |
| Particle images after 2D classification | 1,324,330 | 1,140,275 | 1,311,526 | 46,277 |
| Map sharpening B factor | −117 | −105 | −131 | −490 |

| | P116<br>(PDB 8A9A) | P116 empty<br>(PDB 8A9B) |
|---|---|---|
| **Refinement** | | |
| Initial model used (PDB code) | / | 8A9A |
| Model resolution (Å) | 3.3 | 4 |
| FSC threshold | 0.143 | 0.143 |
| Model resolution range (Å) | 2.5–5.0 | 3.5–6.0 |
| Map sharpening $B$ factor ($Å^2$) | −94 | −105 |
| Model composition | | |
| Non-hydrogen atoms | 12,772 | 6,386 |
| Protein residues | 1,618 | 809 |
| Ligands | 0 | 0 |
| $B$ factors ($Å^2$) | | |
| Protein | 51.22 | 35.30 |
| Ligand | 0 | |
| R.m.s. deviations | | |
| Bond lengths (Å) | 0.003 | 0.027 |
| Bond angles (°) | 0.651 | 1.197 |
| **Validation** | | |
| MolProbity score | 2.18 | 2.52 |
| Clashscore | 12 | 20 |
| Poor rotamers (%) | 0.14 | 0.71 |
| Ramachandran plot | | |
| Favored (%) | 89.34 | 84 |
| Allowed (%) | 10.53 | 15 |
| Disallowed (%) | 0.13 | 1 |

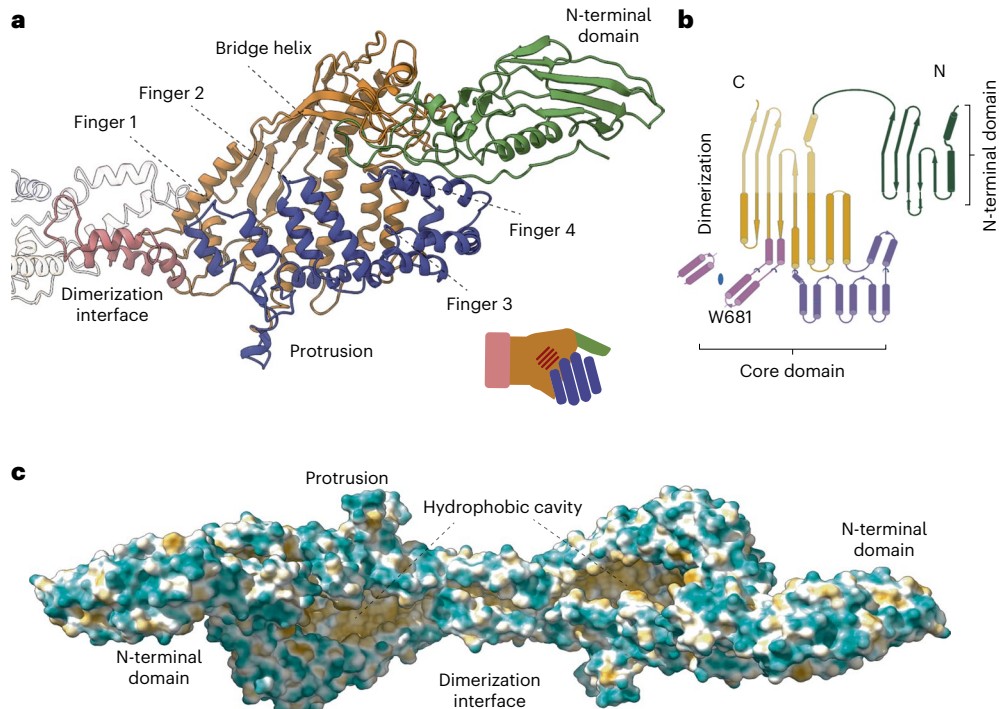

**Fig. 2 | P116 structure and hydrophobic areas. a**, Ribbon model of the P116 monomer, built from the density shown in Figure 1b and colored as in Figure 1. The overall shape of the structure corresponds to a left hand, with the four antiparallel amphipathic α-helices representing fingers (shown in blue), and the bridge helix and β-sheet of five antiparallel strands representing the palm (shown in orange). The N-terminal domain, which is very flexible, corresponds to the thumb. The dimerization helices (shown in pink) correspond to the wrist. **b**, The overall topology of P116. The N-terminal and core domains of P116 share a similar topology, which suggests that P116 might have been generated by duplication of an ancestor domain. Colors correspond to **a. c**, The hydrophobic map of the P116 homodimer shows that the cavity in the core domain is hydrophobic (amino acid hydrophobicity is colored according to the Kyte–Doolittle scale).

Movies 1–3). This distribution contrasts with that of P1, an adhesion protein that concentrates at the tip of the cell and whose inhibition has strong effects on adhesion and motility[16,17].

### P116 has a previously uncharacterized fold with a lipid-accessible cavity

The structure of P116 (30–957) was determined by single-particle cryoEM at 3.3-Å resolution (according to the gold-standard criterion of Fourier shell correlation (FSC) = 0.143; Table 1 and Extended Data Fig. 2). It is an elongated homodimer of ~240 Å along its longest axis, which adopts an arched shape with an arc diameter of ~20 nm (Fig. 1b and Supplementary Movies 4 and 5). Each monomer consists of two distinct domains: the core domain (residues 246–867) and the amino-terminal domain (residues 60–245), situated distal to the dimer axis. The dimerization interface, part of the core domain and proximal to the dimer axis (Fig. 1b and Extended Data Fig. 3a,b), is very well resolved. By contrast, the N-terminal domain has substantial hinge mobility with respect to the core domain, evident by the poorer local resolution of the cryoEM map (Extended Data Fig. 2), making model building difficult for the most distal parts of the construct (see Methods and Extended Data Fig. 3c). The homodimer displays substantial flexibility with many vibrational modes, as illustrated by a complete vibrational analysis, showing a fluent transition between states (Extended Data Fig. 4).

The core domain resembles a half-opened left hand, with four contiguous antiparallel pairs of amphipathic α-helices corresponding to the four fingers and the N-terminal domain corresponding to the thumb (Fig. 2a). The dimer interface, which corresponds to the wrist, is composed of helices with a conserved tryptophan residue (Trp681) that interacts tightly with the neighboring monomer. In the variant with the W681A mutation, the rate of dimers to monomers is

1:4, compared with only dimers in the wild type (Extended Data Fig. 3b). The palm of the hand includes a long and well-defined central α-helix, namely the bridge helix (residues 268–304), and a rigid β-sheet of five antiparallel strands that extends to the N-terminal domain (Fig. 2b). The hand appears in a half-opened state with a large, elongated cleft across the whole core domain (Fig. 2c). The core domain forms a large cavity that measures 62 Å from the proximal to the distal end and 38 Å from the anterior to the posterior side. The cavity has a volume of ~18,000 Å$^3$. The cavity is completely hydrophobic but is fully accessible to the solvent (Fig. 2c and Supplementary Movie 6). In addition, the core has two access points, one at the dorsal side and another at the distal side (Fig. 3a). Using the DALI server, we found only very weak structural relationships between P116 and all other experimentally determined protein structures in the Protein Data Bank, which shows that P116 has a unique fold.

The N-terminal domain is compact and organized around a cluster of aromatic residues, at the center of which is the only tryptophan residue of the domain (W121). The N-terminal and core domains of P116 superimpose for 126 equivalent residues (68% of the N-terminal domain), suggesting that P116 might have been generated by duplication of an ancestor domain. The common secondary structural elements in the N-terminal and core domains consist of a β-sheet and the two helices preceding the β-sheet (Fig. 2b). The core domain is much larger than the N-terminal domain, mainly owing to two insertions containing 12 and 4 helices.

For the inner part of the P116 core domain, the cryoEM maps show prominent elongated densities (with a length of 10–19 Å and a width of 4 Å) that fill most of the hydrophobic areas (Fig. 3a and Supplementary Movies 7 and 8). These elongated densities, which are unaccounted for by the structure, cannot be explained by the protein residues missing in

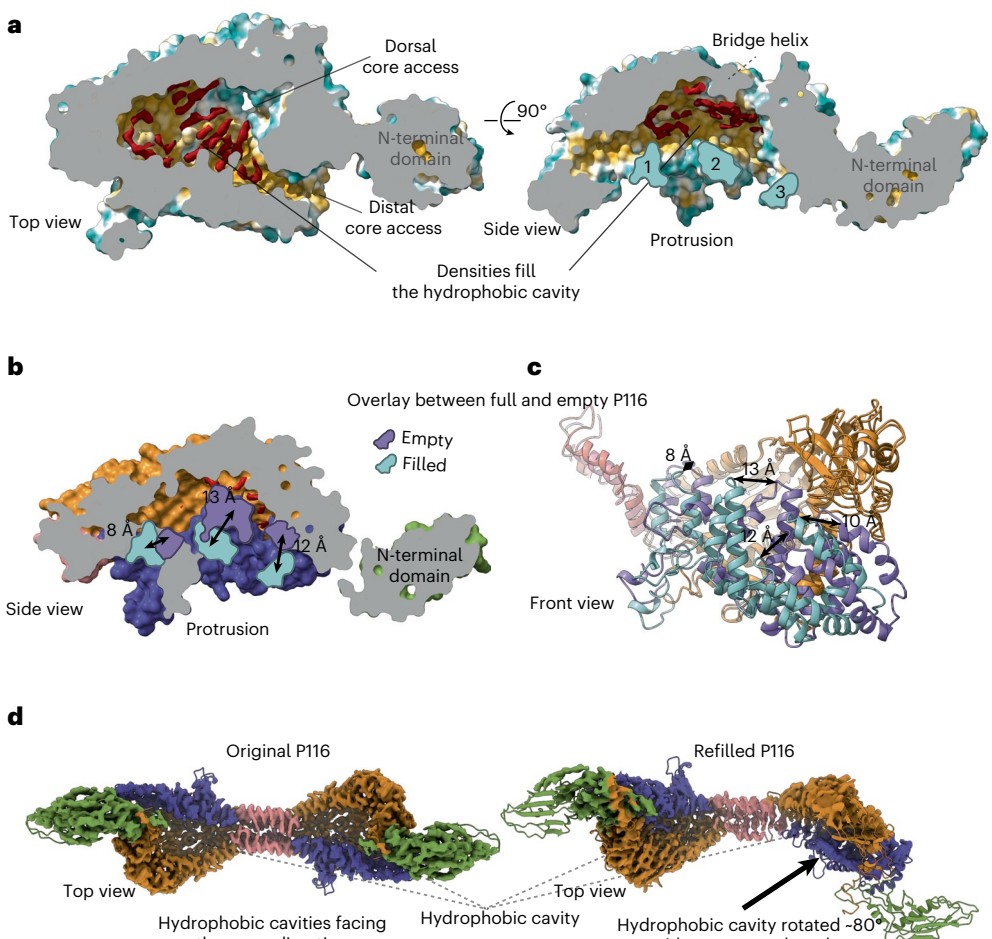

**Fig. 3 | Purified P116 is filled with ligands and displays a large conformational variation compared to empty P116. a**, A cross-section through the core domain of original P116 exposes a series of elongated densities (shown in red), which cannot be accounted for by the structure. These densities are ~4-Å wide and 10- to 19-Å long and are surrounded by highly conserved hydrophobic residues. The cross-section also reveals that the core domain can be accessed dorsally and distally. The side view of the core domain shows that the densities are aligned to the bridge helix and away from the fingers (shown in red). Numbers indicate individual fingers (finger 4 is not visible in this illustration). **b**, Overlay between empty and full P116. The side view of the cross-section surface view of the empty, and full P116 (palm areas aligned) shows that the fingers of the empty P116 (in purple) have come closer to the palm, massively reducing the cavity. The position of the fingers in the empty P116 (in purple) is markedly different

compared with the full P116 (shown in cyan). Finger 1 has moved 8 Å sideways and towards the palm, finger 2 has moved 13 Å towards the palm and finger 3 has moved 12 Å towards the palm. The cavity in the empty P116 is no longer sufficient to accommodate ligands. **c**, In the ribbon presentation, the conformation differences between the empty and full P116 structures can be seen in the front view. All four fingers have moved towards the palm (shown in orange) of the hand (individual distances are indicated filled conformation in cyan, empty conformation in purple). **d**, Two cryoEM classes reveal a wringing movement of P116. Comparison of the two density maps (superimposed with the ribbon diagram of the structure) shows that the wringing movement of P116 allows for the two hydrophobic cavities in the dimer to face almost opposite directions. The top view on the left shows both cavities facing in one direction, and the top view on the right shows the cavities rotated ~80° to each other.

the model. Instead, the mass excess of ~13 kDa, consistently measured by multiple angle light scattering (MALS) and mass spectrometry for P116 in different preparations, could be explained by the presence of ligand molecules bound to P116 (Fig. 4a). Mass spectrometry analysis of the same samples from which the structure of P116 was determined (see Methods) showed the presence of several lipid species, predominantly phosphatidylethanolamine (PE) and phosphatidylglycerol (PG) lipids, as well as wax esters (Fig. 4b–d and Extended Data Fig. 5).

## P116 orthologs in other *Mycoplasma* spp

P116 orthologs were found in at least eight other *Mycoplasma* species, including *M. genitalium* and *M. gallisepticum*. The amino acids lining the hydrophobic cavity are largely conserved (they are either identical or have similar characteristics) (Extended Data Fig. 6a). Modeling the orthologs of P116 with AlphaFold[18] resulted in all the models having a similar tertiary structure, in which a large core domain is flanked by a

smaller N-terminal domain, but the relative position of the domains does not closely match the experimental structure (Extended Data Fig. 6b).

## Empty P116 cannot accommodate lipids

To obtain empty P116 that is free of any bound ligands, we treated the P116 samples with the detergent Triton X-100 (see below and Methods). Mass spectrometry confirmed a massive reduction of lipids in the sample (Fig. 4b). The structure of the empty P116 sample was solved by cryoEM at 4-Å resolution (Extended Data Fig. 7). Its overall topology is almost identical to that of the original P116 sample; however, the core domain is constricted as a result of fingers 1, 2 and 3 being closer to the palm by 8, 13 and 12 Å, respectively, and finger 4 moving 11 Å sideways to retain the distal core access to the cavity (Fig. 3b, Supplementary Movies 9 and 10 and Extended Data Fig. 8). These changes reduce the volume of the cavity from ~18,000 Å³ to

~6,300 Å³. Consequently, the huge hydrophobic cavity reduces to two pockets that are large enough for lipids to pass through but that appear unoccupied in the cryoEM density. A comparison of the filled and empty P116 structures shows that the original densities that were unaccounted for create massive steric clashes in the constricted configuration, demonstrating that the cavity can no longer accommodate lipids (Supplementary Movie 11). In the empty P116, the dimerization interface is shifted towards the dorsal side of the molecule by 10 Å, resulting in a contraction that changes the arc diameter of the dimer to ~10 nm and shifts the N-terminal domain towards the dimerization interface.

### Refilled P116 is identical to the purified P116

We next refilled the empty P116 samples by incubating them either with fetal bovine serum (FBS) or with high-density lipoprotein (HDL) and then re-purified them by affinity chromatography. Medium containing FBS is a common growing broth for *M. pneumoniae* cultures, although lipoproteins, in particular HDL, are efficient substitutes for serum in mycoplasma culture medium, likely because lipoproteins can provide the key lipids, in particular cholesterol, which are essential for mycoplasma cells[19]. We solved the structure of the refilled P116 samples at 3.5-Å resolution using cryoEM. The structure of the refilled monomer P116 is practically identical to the structure of the original monomer P116 sample, including densities at the palm of the hand that can be assigned to ligands. Mass spectrometry of the refilled samples shows the clear presence of lipids (Fig. 4b).

### Conformational flexibility of P116

The structures of the original P116, empty P116 and refilled P116 samples appear predominantly as homodimers. In all cases, the homodimer exhibits substantial flexibility. Most prominently, the arc diameter of the empty structure is approximately 10 nm smaller than that of the original and refilled structures. In addition, a wringing motion is visible in the refilled structure: each monomer is twisted in the opposite direction along the axis perpendicular to the dimer axis by ~80°, and bends up to 20°, depending on its cargo (Fig. 3d, Supplementary Movie 12 and Extended Data Fig. 4). In all P116 structures, the N-terminal domain is the most flexible. Within the core domain, temperature factors are higher at the fingertips, indicating movement of the antiparallel α-helices. When the fingers approach the palm, this results in the core domain constricting and a clash with the densities therein (Supplementary Movie 11).

### P116 ligands include essential lipids

We next set out to characterize the possible ligands within P116. We first measured the rate of radioactivity transfer to P116 after incubation with HDL particles containing either tritium-labeled cholesterol ([³H]cholesterol) or tritium-labeled cholesteryl oleate as a representative of cholesterol esters (Table 2). A substantial fraction of the [³H]

cholesterol-containing HDL radiotracer was detected in the P116 samples that had been incubated with HDL and then separated from it by purification (Methods and Extended Data Fig. 9), indicating a net transfer of both cholesterol and cholesterol ester between HDL and P116. The total absence of the most abundant HDL protein (APOA1), cross-checked by immune detection (Methods and Supplementary Table 2), verified that no HDL remnants had contaminated the purified P116 samples.

The highest rate of radiotracer transfer was achieved when [³H] cholesterol-containing HDLs were mixed with empty P116. [³H]cholesterol was also transferred. Transfer of [³H]cholesterol esters to P116 would require a direct interaction between HDL and P116, as these esters are buried in the core of the HDL particles (Table 2). Passive cholesterol transport from cellular membranes to HDL or from low-density lipoprotein to HDL has been reported[20], but the concept that bacteria can actively extract cholesterol from HDL has not been previously characterized.

We then conducted a detailed matrix-assisted laser desorption/ ionization–time of flight (MALDI-TOF) and liquid chromatography electrospray ionization coupled with tandem mass spectrometry (LC-ESI–MS/MS) analysis. We identified more than 500 lipid species in the samples and found striking differences between the original, empty and refilled P116 samples (Fig. 4b–d). In the original P116 sample, the predominant lipid species were PE, PG and wax esters. Wax esters are not known to be required by *M. pneumoniae*, but they were part of the cultivation medium of the *E. coli* strain in which P116 was produced. Incorporation of many lipid species is in agreement with the fact that *M. pneumoniae* adapts its membrane composition to the available lipid spectrum[6–8]. In the empty P116, we observed a substantial reduction of lipids, with no specific lipid class enriched. In the P116 samples refilled from FBS, we observed a clear accumulation of the essential lipid classes phosphatidylcholine (PC) and sphingomyelin (SM), as well as sterols and cholesterol (Fig. 4c,d and Supplementary Table 3).

### P116 extracts specific lipid classes

To analyze lipoprotein carryover in the FBS-refilled P116, we conducted an additional proteomics LC–MS/MS experiment (Supplementary Table 4) using ultrasensitive, ion-mobility-assisted LC–MS/MS. In this experiment, we observed limited lipoprotein carryover into the refilled sample. However, on the basis of peptide spectrum match (PSM) numbers and intensity values, we found P116 to be over 30-fold more abundant than the lipoproteins in the refilled sample. If the lipid spectrum in the FBS-refilled P116 sample originated from lipoprotein carryover, we would expect a similar distribution of the lipid classes in both samples. In fact, we observed a specific enrichment of PC and SM in the FBS-refilled sample, whereas TG, the most abundant lipid class in the serum, was decreased and was barely detectable. Thus, although P116 can extract a large range of lipids, it shows a preference for selected

**Fig. 4 | Analysis of the lipid spectrum and uptake of P116. a**, MALDI-TOF mass spectrum of original P116 sample (linear mode, high mass range), showing a dominant peak at 105 kDa, corresponding to the singly charged full protein, as well as the charged states two, three and four. a.u., arbitrary units. **b**, Stacked MALDI-TOF mass spectra (reflector mode, low mass range) of the original purified P116 (purple, rear), empty P116 (black, middle) and refilled P116 sample (orange, front) showing a change in the lipid distribution among the samples. **c,d**, Hierarchical clustering of lipid compounds identified in positive (**c**) and negative (**d**) ion mode lipidomics (LC–MS/MS) analyses (reproduced in three independent experiments), showing differential distributions of lipid compositions in original P116 (first column), empty P116 (second column), refilled P116 (third column) and serum (fourth column). The refilled P116 shows a particular affinity to sterols and cholesterol specifically. All data were normalized to the mTIC of all identified compounds in each sample, and row-wise scaling was applied. PE, phosphatidylethanolamine; PG, phosphatidylglycerol; DG, diacylglycerol; PC, phosphatidylcholine; SM, sphingomyelin; TG, triacylglycerol; FA, fatty acid; LPC, lysophosphatidylcholine; VAE, vitamin

A fatty acid ester; SE, sterol esters; PI, phosphatidylinositol; NAE, N-acyl ethanolamines; LPE, lysophosphatidylethanolamine; LDGTS, lysodiacylglyceryl trimethylhomoserine; DGGA, diacylglyceryl glucuronide; CE, cholesteryl ester; BMP, bismonoacylglycerophosphate; NAE, N-acyl ethanolamines; MGDG, monogalactosyldiacylglycerol; HBMP, hemibismonoacylglycerophosphate; DGTS, diacylglyceryl trimethylhomoserine. **e**, CryoEM analysis of empty P116 incubated with HDL shows that P116 binds HDLs between its N-terminal and core domains. P116 is attached to HDL through its distal core. Owing to the flexibility of P116 and the variability of HDL, only one subunit of P116 can be seen at this threshold. The whole P116 can be seen in the individual class averages. **f**, Schematic of the lipid uptake and conformational variations of P116 (here indicated by its structure anchored in the mycoplasma membrane. Linkers and transmembrane domains not seen in the cryoEM structure are shown in purple). P116 starts in an empty, constricted state; incubation with HDL leads to each individual monomer filling up with approximately 20 lipids; and P116 changes to the open/filled state. We hypothesize that, through a wringing motion, lipids are delivered into the mycoplasma membrane.

lipid species (Fig. 4c,d and Supplementary Table 3). We conclude that the lipid composition in the FBS-refilled P116 sample can be attributed predominantly to P116 itself and not to lipoprotein carryover.

## P116 binds at defined regions to HDL

Next, we performed cryoEM on a sample containing empty P116 and HDL. Of ~58,000 particles that were identified as HDL, ~25,000 were attached to P116. The resulting density at a resolution of 9 Å shows P116

interacting directly with HDL. The structure can be well fitted to the density map. Interestingly, the P116 region between the N-terminal domain and the core interacts with HDL (Fig. 4f). Cryo-electron tomograms of whole *M. pneumoniae* cells indicate that this region faces away from the *M. pneumoniae* membrane and is thus accessible to vesicles and lipids. This presents a possible explanation as to how P116 avoids extracting lipids from the *M. pneumoniae* membrane itself. However, the unambiguous identification of P116 on the *M. pneumoniae* membrane is

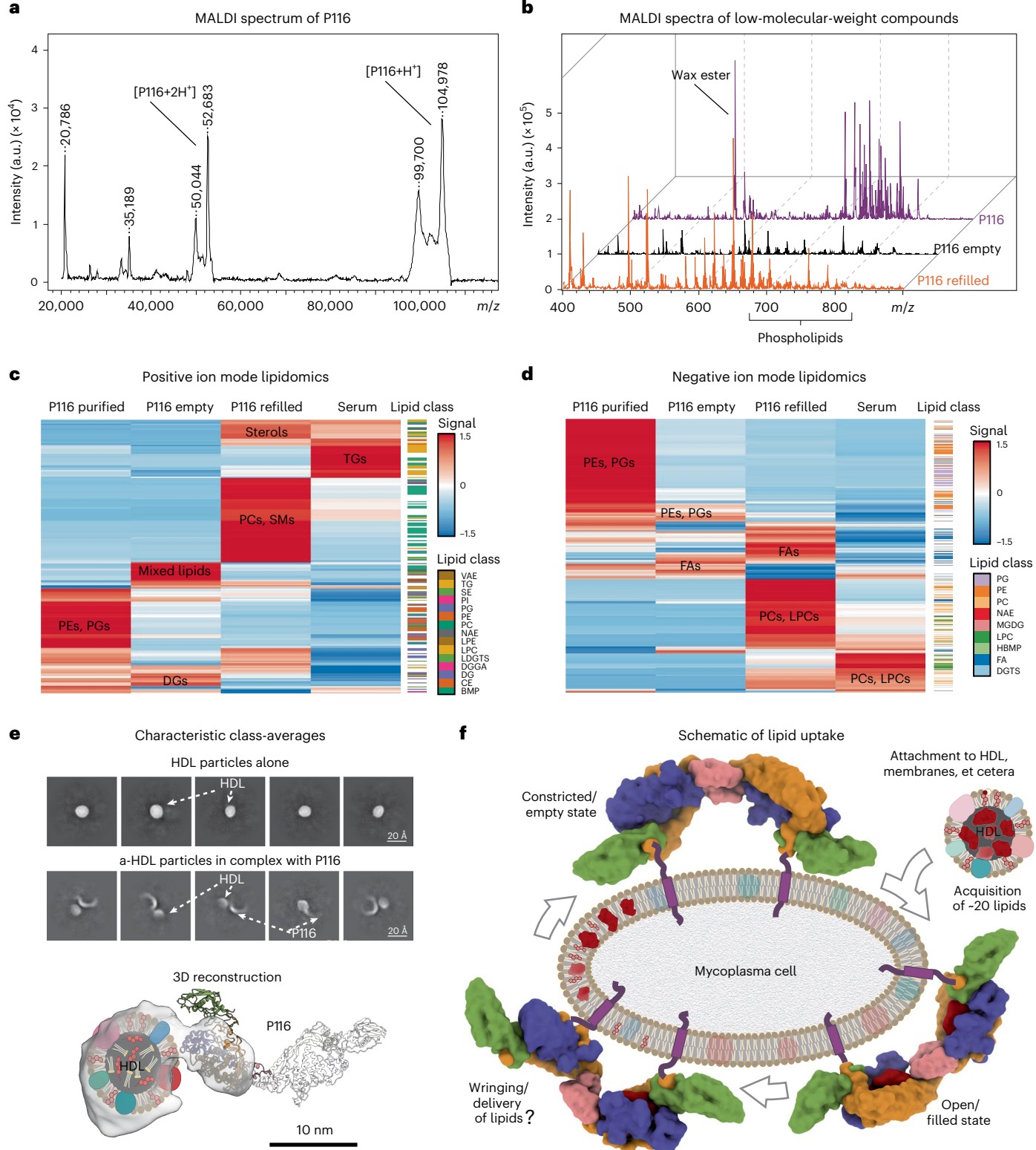

**Table 2 | Relative transfer of cholesterol from HDL to P116**

| | % of [³H] cholesterol transferred/mL | nmol, cholesterol transferred/ mL/hour | nmol cholesterol transferred/mg P116[a] |
|---|---|---|---|
| **HDL to empty P116** | | | |
| Free cholesterol | 13.12 | 13.52 | 59.49 (6.3) |
| Esterified cholesterol | 6.98 | 7.22 | 31.75 (3.3) |
| **HDL to original P116** | | | |
| Free cholesterol | 7.89 | 7.42 | 32.63 (3.4) |
| Esterified cholesterol | 6.32 | 6.01 | 26.44 (2.8) |

[a]Numbers in parentheses are the estimated number of cholesterol molecules transferred per P116 subunit (assuming a molecular weight of ~105 kDa for the construct).

challenging owing to the low copy number of P116 (ref. [14]), and further experiments are required to better characterize the attachment of P116 with the *M. pneumoniae* membrane (Extended Data Fig. 10).

## Discussion

P116 is essential for the viability of the human pathogen *M. pneumoniae*[4] and is the target of a strong antigenic response in infected people[21]. The P116 structure has a previously unseen fold with a uniquely large hydrophobic cavity filled with ligands. Mass spectrometry and radioactivity transfer experiments confirm a lipid extraction from serum (FBS) and HDL. Further, the ligands have been identified as essential lipids for the survival of the cells. In fact, we found a high specificity towards cholesterol, PC and SM, which are the most abundant membrane lipids in *M. pneumoniae*[8]. Crosslinking mass spectrometry studies indicate one weak amino acid-pair interaction between P116 and MPN161 (a protein of unknown function)[22]. Thus, although the involvement of other proteins in incorporating the extracted lipids into the *Mycoplasma* membrane cannot be excluded, it appears likely, given the observed conformational states upon lipid extraction, that P116 is also responsible for incorporation, thus P116 is responsible for the complete uptake (Fig. 4f). Taken together, the P116 structure and our insights into different P116 conformations and the P116 complex formation with HDL reveal a mechanism by which *Mycoplasma species* extract lipids from the environment and most likely incorporate them into their own membrane.

The transition from a full to an empty P116 molecule involves a ~70% volume reduction of the hydrophobic cavity in concert with a wringing motion of the core domains. During this wringing motion, in which the monomers are each twisted in the opposite direction around their long axis, the hydrophobic cavities face almost opposite directions. Because the N-terminal domain is near the C terminus, which anchors the protein in the *Mycoplasma* membrane in vivo, the core is the domain that experiences the high flexibility seen in our data sets. This flexibility enables an alternating wringing motion whereby one monomer of the core domain faces the *Mycoplasma* membrane (that is, the monomer transferring lipids to the membrane) and the other monomer faces the environment (that is, the monomer extracting lipids from the environment). This wringing motion can be repeated in a continuous manner. In this way, P116 could undergo a rolling movement on the *Mycoplasma* membrane, thus facilitating the transport of cholesterol and other essential lipids in an apparently simple way for lipid transporters (Fig. 4f).

*Mycoplasma* species have a minimal genome and are capable of incorporating many different lipids into their membranes[6,7]. The lipid-binding versatility shown by P116 enables a single molecular system to cope with the transport of diverse lipids required by *Mycoplasma*. Although only *Mycoplasma* shares genes with sequences similar to that of *p116*, other microorganisms that require uptake of lipids from the environment, including clinically relevant bacterial species such as *B. burgdorferi*, may have similar, as yet undiscovered systems to regulate their cholesterol homeostasis. Whether P116 shares functional similarities with other transfer proteins such as human cholesteryl ester transfer and phospholipid transfer proteins[23,24] requires further investigation. However, the diversity and amount of lipids that P116 can bind appear to be unmatched by any other known prokaryotic or eukaryotic lipid carrier. Interestingly, despite its broad lipid range, P116 still shows a high specificity, largely enriching certain lipids (SM, PC and cholesterol) while excluding others (TGs). This understanding of bacterial lipid uptake presents potential opportunities for treatment of mycoplasma infections and may for the first time[2] enable the development of a vaccine against *M. pneumoniae*.

## Online content

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

## Methods

The research complies with all relevant ethical regulations. The experimental procedures to immunize mice and obtain monoclonal antibodies were approved by the Ethics Committee on Animal and Human Experimentation from the Universitat Autònoma de Barcelona under the document CEEAH 1002R3R2R.

### Bacterial strains, tissue cultures and growth conditions

The *M. pneumoniae* M129 strain was grown in cell culture flasks containing SP4 medium and incubated at 37 °C and 5% $CO_2$. Surface-attached mycoplasmas were collected using a cell scraper and resuspended in SP4 medium. To grow mycoplasma cells on IBIDI eight-well chamber slides, each well was seeded with about $1 \times 10^5$ colony-forming units and incubated for 12–24 hours in 200 µL SP4 supplemented with 3% gelatin.

NSI myeloma cells[25] were grown in RPMI 1640 medium supplemented with 10% fetal bovine serum (FBS) and 50 µg mL$^{-1}$ gentamicin (complete RPMI). Hybridomas were selected in complete RPMI supplemented with HAT medium and BM-Condimed (Sigma Aldrich).

### Cloning, expression and purification of P116 constructs

Regions corresponding to the *mpn_213* gene from *M. pneumoniae* were amplified from synthetic clones using different primers for each construct: P116F$_{30}$ and P116R$_{957}$ for P116(30–957); P116F$_{13}$ and P116R$_{957}$ for P116(13–957); P116F$_{212}$ and P116R$_{862}$ for P116(212–862); and P116W$_{681}$ to generate variant P116 W681A. PCR fragments were cloned into the expression vector pOPINE (gift from R. Owens; plasmid no. 26043, Addgene) to generate constructs, with a carboxy-terminal His-tag. Recombinant proteins were obtained after expression at 22 °C in B834 (DE3) cells (Merck), upon induction with 0.6 mM IPTG at an optical density at 600 nm (OD$_{600}$) of 0.8. Cells were collected and lysed by French press in binding buffer (20 mM TRIS-HCl pH: 7.4, 40 mM imidazole and 150 mM NaCl) and centrifuged at 49,000*g* at 4 °C. Supernatant was loaded onto a HisTrap 5 ml column (GE Healthcare) that was pre-equilibrated in binding buffer and elution buffer (20 mM TRIS-HCl pH: 7.4, 400 mM imidazole and 150 mM NaCl). Soluble aliquots were pooled and loaded onto a Superdex 200 GL 10/300 column (GE Healthcare) in a protein buffer (20 mM TRIS-HCl pH 7.4 and 150 mM NaCl).

To obtain empty P116, 2.6% Triton X-100 was added to the protein sample and incubated for 1.5 hours at room temperature. Subsequent purification followed the same methodology described above, but also included a wash step with the binding buffer plus 1.3% Triton X-100, followed by extensive washing with at least 20 column volumes of wash buffer (20 mM TRIS-HCl pH: 7.4, 20 mM imidazole) before eluting the samples from the column. P116 was concentrated with Vivaspin 500 centrifugal concentrators (10,000 MWCO PES, Sartorius) to a final concentration of >0.5 mg/mL.

To refill P116 with lipids, the empty protein was incubated with approximately 1 mL FBS per mg P116 for 2 hours at 30 °C while still bound on the column. After extensive washing with at least 40 column volumes of wash buffer, elution and concentration were performed as described above.

### HDL isolation and determination of cholesterol transfer rate

Human HDL (density 1.063–1.210 g/mL) was isolated from plasma of healthy donors through sequential gradient density ultracentrifugation, using potassium bromide for density adjustment, at 100,000*g* for 24 hours with an analytical fixed-angle rotor (50.3, Beckman Coulter). The amount of cholesterol and apolipoprotein A1 were determined enzymatically and by an immunoturbidimetric assay, respectively, using commercial kits adapted for a COBAS 6000 autoanalyzer (Roche Diagnostics, Rotkreuz, Switzerland). Radiolabeled HDLs were prepared in the following way: 10 µCi of either [1,2-³H(N)] free cholesterol or [1,2-³H(N)]cholesteryl oleate (Perkin Elmer) were mixed with absolute ethanol, and the solvent was dried under a stream of $N_2$. HDL (0.5 mL, 2.25 g/L of ApoA1) was added to the tubes containing the

radiotracers, as appropriate, and then incubated for 16 hours in a 37 °C bath[26]. The labeled HDLs (both ³H-cholesterol-containing and ³H-cholesteryl oleate-containing HDLs) were re-isolated by gradient density ultracentrifugation at 1.063–1.210 g/mL and dialyzed against PBS through gel filtration chromatography. Specific activities of ³H-cholesterol-containing and ³H-cholesteryl oleate-containing HDLs were 1,221 and 185 counts per minute (cpm)/nmol, respectively. The cholesterol transfer to P116 (1 g/L) was measured after adding either [³H] free cholesterol-containing or [³H]cholesteryl oleate-containing HDL (0.5 g/L of APOA1) and incubation for 2 hours at 37 °C. HDL and P116 were separated by a HisTrap HP affinity and size-exclusion columns (Extended Data Fig. 10). The radioactivity associated with each P116 and HDL fraction was measured through liquid scintillation counting. The percentage of [³H]cholesterol transferred per mL was determined for each condition. The specific activities for each radiotracer were used to calculate the amount of free cholesterol and cholesteryl ester transferred from HDL to P116. Total cholesterol levels in the HDL fraction were determined enzymatically by using a commercial kit adapted for a COBAS 6000 autoanalyzer (ref 03039773190, Roche Diagnostics). Human APOA1 levels were determined in both the HDL and purified P116 fractions by an assay (ref 3032566122, Roche Diagnostics) that used anti-APOA1 antibodies that react with the antigen in the sample to form antigen–antibody complexes, which, after agglutination, were measured turbidimetrically in the COBAS 6000 autoanalyzer (Supplementary Table 2).

### Size-exclusion chromatography and multi-angle light scattering

Molecular weights were measured from P116 samples using a Superose 6 10/300 GL (GE Healthcare) column in a Prominence liquid chromatography system (Shimadzu) connected to a DAWN HELEOS II multi-angle light scattering (MALS) detector and an Optilab T-REX refractive index (dRI) detector (Wyatt Technology). ASTRA 7 software (Wyatt Technology) was used for data processing and analysis. An increment of the specific refractive index in relation to concentration changes (dn/dc) of 0.185 mL/g (typical of proteins) was assumed for calculations.

### Matrix-assisted laser desorption/ionization-mass spectrometry

All samples were mixed in a 1:1 ratio with sDHB (Super-DHB, Bruker) matrix solution (50 mg mL in 50% acetonitrile (ACN), 50% water, and 0.1% trifluoroacetic acid). Subsequently, 1-µL aliquots of the mixture were deposited on a BigAnchor MALDI target (Bruker) and allowed to dry and crystallize at ambient conditions.

MS spectra were acquired on a rapifleX MALDI-TOF/TOF (Bruker, Germany) in the mass range of 20,000–120,000 *m/z* in linear positive mode for intact protein measurements and in the mass range of 100–1,600 *m/z* in reflector positive mode for lipid measurements. The Compass 2.0 (Bruker) software suite was used for spectra acquisition and processing.

### Lipidomics analysis (LC–MS/MS)

Lipid samples, with an equivalent of 10 µg of protein, were extracted using a modified MTBE/Methanol extraction protocol[27] and submitted to LC–MS/MS analysis using a nanoElute (Brukrt) system, equipped with C18 analytical column (15 cm × 75 µm, particle size: 1.9 µm (PepSep)), coupled to a timsTOF Pro 2 mass spectrometer (Bruker).

Samples were loaded directly onto the analytical column with twice the sample pick-up volume with buffer A. Lipids were separated on the analytical column at 60 °C with a flow rate of 400 nL/minute, with the following gradient: 1% B for 1 minute, 1 to 30% B in 2 minute, 30 to 51% B in 4 minutes, 51 to 61% B in 5 minutes, 61 to 70% B in 5 minutes, 70 to 99% B in 5 minutes and constant 99% B for 13 minutes. This was followed by column re-equilibration with buffer A (ACN/water (60/40, vol/vol) with 10 mM ammonium formate and 0.1% FA) and

buffer B (2-propanol/ACN (90/10, vol/vol) with 10 mM ammonium formate and 0.1% FA).

Lipids eluting from the column were ionized online using a captive spray ion source and were analyzed in two replicates for positive and negative mode using DDA-PASEF with a ramp time of 100 ms and 3 PASEF-MS/MS events. Spectra were acquired over the mass range from 50–1,550 $m/z$ and a mobility window from 0.55–1.95 Vs/cm$^2$.

Raw data were converted into ibf files and analyzed using the MS-DIAL lipidomics pipeline (version 4.9 (ref. [28])) with default processing parameters for timsTOF data. Identified lipids were aligned to a pooled control sample and filtered by blank abundances (sample intensity/blank intensity >5). Intensities were normalized by mTIC, exported and further analyzed in R using the lipidr package[29].

### Proteomics analysis (LC–MS/MS)
Protein samples were reduced with TCEP and cysteines alkylated with IAA (Thermo Fisher). Subsequent proteolytic digests were performed using S-TRAPs (Protifi), according to the manufacturer's instructions. Peptides were further desalted and purified on Isolute C18 SPE cartridges (Biotage, Sweden) and dried in an Eppendorf concentrator (Eppendorf).

After solubilization in 0.1% formic acid (FA) in ACN/water (95/5, vol/vol), samples were subjected to LC–MS/MS analysis on a nanoElute (Bruker) system, equipped with C18 analytical column (15 cm × 75 µm, particle size: 1.9 µm (PepSep)) coupled to a timsTOF Pro 2 mass spectrometer (Bruker).

Samples were loaded directly onto the analytical column with twice the sample pick-up volume with buffer A. Peptides were separated on the analytical column at 60 °C with a flow rate of 500 nL/minute, with the following gradient: 2 to 35% B in 17.8 minutes, 35 to 95% B in 0.5 minutes and constant 90% B for 2.4 minutes with buffer A (0.1% FA in water) and buffer B (0.1% FA in acetonitrile).

Peptides eluting from the column were ionized online using a captive spray ion-source and analyzed in DDA-PASEF mode with a cycle time of 100 ms and 4 PASEF-MS/MS events. Spectra were acquired over the mass range of 100–1,700 $m/z$ and a mobility window of 0.85–1.3 Vs/cm$^2$

Data analysis was performed in FragPipe 18 using MSFragger 3.5 for database searches[30]. Raw files were recalibrated, search parameters automatically optimized and searched against the combined Uniprot reference proteomes for *M. pneumoniae, E. coli* and *Bos taurus* (UP000000808, UP000000625, UP000162055; obtained 2022-06-23).

The database search space was restricted to tryptic peptides with a length of 7–50 amino acids, allowing for up to two missed cleavages and with a minimum of one unique peptide per protein group. Carbamidomethylation of cysteine was set as a fixed modification and oxidation of methionine, as well as N-terminal acetylation, were set as variable modifications. Percolator was used to estimate the number of false positive identifications, and the results were filtered for a strict target false discovery rate (FDR) < 0.01.

### Single-particle cryoEM
For single-particle cryoEM, a 3.5-µL drop of purified P116 (100–400 µg/mL in 20 mM Tris, pH 7.4 buffer or 600 µg/mL in 20 mM Tris, 2 mM CHAPSO, pH 7.4 buffer) or P116 mixed with HDL (250 µg/mL P116 and 1116 µg/mL HDL in 20 mM Tris, pH 7.4 buffer) was applied to a 45 s glow-discharged R1.2/1.3 C-flat grid (Electron Microscopy Science), and plunge-frozen in liquid ethane (Vitrobot Mark IV, Thermo Scientific) at 100% relative humidity, 4 °C, a nominal blot force of −3, and a wait time of 0.5 seconds, with a blotting time of 12 s. Before freezing, Whatman 595 filter papers were incubated for 1 hour in the Vitrobot chamber at 100% relative humidity and 4 °C.

Dose-fractionated movies of P116, P116 refilled and P116 mixed with HDL were collected with SerialEM v3.8 (ref. [31]) at a nominal magnification of ×130,000 (1.05 Å per pixel) in nanoprobe EFTEM mode at 300 kV with a Titan Krios (Thermo Scientific) electron microscope

equipped with a GIF Quantum S.E. post-column energy filter in zero loss peak mode and a K2 Summit detector (Gatan). For P116, P116 refilled and P116 with HDL, a total of 4,376, 4,019 and 3,114 micrographs with 34, 29 and 30 frames per micrograph and a frame time of 0.2 seconds were collected. The camera was operated in dose-fractionation counting mode with a dose rate of ~8 electrons per Å$^2$ s$^{-1}$, resulting in a total dose of 50 electrons per Å$^2$ s$^{-1}$. Defocus values ranged from −1 to −3.5 µm.

For P116 empty, dose-fractionated movies were collected using EPU 2.12 (Thermo Scientific) at a nominal magnification of ×105,000 (0.831 Å per pixel) in nanoprobe EFTEM mode at 300 kV with a Titan Krios G2 electron microscope (Thermo Scientific), equipped with a BioQuantum-K3 imaging filter (Gatan), operated in zero loss peak mode with 20 eV energy slit width. In total,15,299 micrographs with 50 frames per micrograph and frame time of 0.052 seconds were collected. The K3 camera was operated in counting mode with a dose rate of ~16 electrons per A$^2$ s$^{-1}$, resulting in a total dose of 50 electrons per Å$^2$ s$^{-1}$. Defocus values ranged from −0.8 to −3.5 µm.

CryoSPARC v3.2 (ref. [32]) was used to process the cryoEM data, unless stated otherwise. Beam-induced motion correction and CTF estimation were performed using CryoSPARC's own implementation. Particles were initially clicked with the Blob picker using a particle diameter of 200–300 Å. Particles were then subjected to unsupervised 2D classification. For the final processing, the generated 2D averages were taken as templates for the automated particle picking; for the processing of P116 with HDL, no template picking was performed. In total, 3,463,490, 4,532,601 particles, 2,930,863 particles and 262,981 particles were picked and extracted with a binned box size of 256 pixels for P116, P116 empty, P116 refilled and P116 with HDL, respectively. False-positive picks were removed by two rounds of unsupervised two-dimensional classification. The remaining 1,324,330 particles (P116), 1,140,275 particles (P116 empty), 1,311,526 particles (P116 refilled) and 46,277 particles (P116 with HDL) were used to generate an ab initio reconstruction with three classes followed by a subsequent heterogeneous refinement with three classes. For the final processing, 1,315,362 particles (P116), 633,332 particles (P116 empty), 1,311,526 particles (P116 refilled) and 46,277 particles (P116 with HDL) were used. For the remaining particles, the beam-induced specimen movement was corrected locally.

The CTF was refined per group on the fly within the non-uniform refinement. The obtained global resolution of the homodimer was 3.3 Å (P116), 4 Å (P116 empty), and 3.5 Å (P116 refilled) (Extended Data Figs. 2 and 8 and Table 1). To analyze the sample with regard to its flexibility, the particles were subjected to the 3D variability analysis of cryoSPARC, which was used to display the continuous movements of the protein.

### Cryo-electron tomography of *M. pneumoniae*
*M. pneumoniae* M129 cells of an adherently growing culture were scraped in a final volume of 1 mL of SP4 medium and washed three times in PBS. This solution was mixed with fiducial markers (Protein A conjugated to 5 nm colloidal gold: Cell biology department, University Medical Center Utrecht). From this stock, a 3.5-µL drop was applied to a (45 s) glow-discharged R1.2/1.3 C-flat grid (Electron Microscopy Science), and plunge-frozen in liquid ethane (Vitrobot Mark IV, Thermo Scientific) at 100% relative humidity, 4 °C, and a nominal blot force of −1, with a wait and blotting time of 10 seconds.

Tilt-series were recorded using SerialEM v3.8 (ref. [31]) at a nominal magnification of ×105,000 (1.3 Å per pixel) in nanoprobe EFTEM mode at 300 kV with a Titan Krios (Thermo Scientific) electron microscope equipped with a GIF Quantum S.E. post-column energy filter in zero loss peak mode and a K2 Summit detector (Gatan). The total dose per tomogram was 120 e$^-$/Å$^2$, and the tilt series covered an angular range from −60° to 60° with an angular increment of 3° and a defocus set at −3 µm. Tomograms were reconstructed by super-sampling SART[33] with a 3D CTF correction[34].

## P116 model building and refinement

The initial tracing of the core domain was performed manually with Coot[35]. It contained numerous gaps and ambiguities that were slowly polished by alternating cycles of refinement using the 'Real Space' protocol in the program Phenix[36,37] and manual reinterpretation and rebuilding with Coot. The tracing and assignment of specific residues in the N-terminal domain were very difficult owing to the low local resolution of the map for this domain, and only a partial interpretation was achieved. Using Robetta and AlphaFold[18], we obtained different predictions of the N-terminal domain structure using different parts of the sequence. The highest ranked predictions, selected using the partial experimental structure already available, were obtained with AlphaFold for residues 81–245, which allowed us to complete the building of the N-terminal domain according to the cryoEM map. The root-mean-square deviation between the AlphaFold prediction and the experimental model was 2.6 Å for 104 (63%) structurally equivalent residues. Some residues at the N end of the N-terminal domain were difficult to identify and were represented as alanines in the final model. The whole P116 model was then refined using Phenix, and the final refined structure was deposited in the EMDB under code EMD-15274 (Table 1).

## Polyclonal and monoclonal antibody generation

Two BALB/C mice were serially immunized with four intraperitoneal injections, each containing 150 μg of recombinant P116 ectodomain (residues 30–957) in 200 μL of PBS with no adjuvants. The last injection was delivered 4 days before splenectomy. Isolated B lymphocytes from the immunized mice were fused to NSI myeloma cells[25] to obtain stable hybridoma cell lines producing monoclonal antibodies[38]. Supernatants from hybridoma cell lines derived from single fused cells were first investigated by indirect ELISA screening against the recombinant P116 ectodomain. Positive clones were also tested by western blot against protein profiles from *M. pneumoniae* cell lysates and by immunofluorescence using whole, non-permeabilized *M. pneumoniae* cells (see below). Only those clones with supernatants revealing a single 116-kDa band in protein profiles and also exhibiting a consistent fluorescent staining of *M. pneumoniae* cells were selected and used in this work. Polyclonal sera were obtained by cardiac puncture of properly euthanized mice just before splenectomy and titred using serial dilutions of the antigen. The titer of each polyclonal serum was determined as the half-maximal inhibitory concentration ($IC_{50}$) value from four parameter logistic plots and was found to be approximately 1/4,000 for both sera. Polyclonal anti-P1 antibodies were obtained by immunizing two BALB/C mice with recombinant P1 proteins[39], respectively, as described above. The titers obtained for polyclonal anti-P1 antibodies were approximately 1/2,500 and 1/3,000, respectively.

## Immunofluorescence microscopy

The immunofluorescence staining of mycoplasma cells on chamber slides was similar to previously described[40], with several modifications. Cells were washed with PBS containing 0.02% Tween 20 (PBS-T) prewarmed at 37 °C, and each well was fixed with 200 μL of 3% paraformaldehyde (wt/vol) and 0.1% glutaraldehyde. Cells were washed three times with PBS-T, and slides were immediately treated with 3% BSA in PBS-T (blocking solution) for 30 minutes. The blocking solution was removed, and each well was incubated for 1 hour with 100 μL of the primary antibodies diluted in blocking solution. For P116 and P1 polyclonal sera, we used a 1/2,000 dilution; a 1/10 dilution was used for monoclonal antibodies from hybridoma supernatants. Wells were washed three times with PBS-T and incubated for 1 hour with a 1/2,000 dilution of a goat anti-mouse Alexa 555 secondary antibody (Invitrogen) in blocking solution. Wells were then washed three times with PBS-T and incubated for 20 minutes with 100 μL of a solution of Hoechst 33342 10 μg/μL in PBS-T. Wells were finally washed once with PBS-T and replenished with 100 μL of PBS before microscopic

examination. Cells were observed by phase contrast and epifluorescence in an Eclipse TE 2000-E inverted microscope (Nikon). Phase contrast images, 4′,6-diamidino-2-phenylindole (DAPI, excitation 387/11 nm, emission 447/60 nm) and Texas Red (excitation 560/20 nm, emission 593/40 nm) epifluorescence images were captured with an Orca Fusion camera (Hamamatsu) controlled by NIS-Elements BR software (Nikon).

## Time-lapse microcinematography

The effect of anti-P116 antibodies and anti-P1 polyclonal serum on mycoplasma cell adhesion was investigated by time-lapse cinematography of *M. pneumoniae* cells growing on IBIDI eight-well chamber slides. Before observation, medium was replaced with PBS containing 10% FBS and 3% gelatin prewarmed at 37 °C. A similar medium has been used to test the effect of P1 antibodies on mycoplasma adhesion and gliding motility[41]. After incubation for 10 minutes at 37 °C and 5% $CO_2$, the slide was placed in a Nikon Eclipse TE 2000-E inverted microscope equipped with a Microscope Cage Incubation System (Okolab) at 37 °C. Images were captured at 0.5-second intervals, for a total observation time of 10 minutes. After the first 60 seconds of observation, the different antibodies were dispensed directly into the wells. The frequencies of motile cells and detached cells before the addition of antibodies were calculated from the images collected between 0 and 60 seconds of observation. The frequencies of motile cells and detached cells after the addition of antibodies were calculated from the images collected in the last minute of observation.

## Reporting summary

Further information on research design is available in the Nature Portfolio Reporting Summary linked to this article.

## Data availability

Cryo-electron microscopy densities of the original P116 density map (3.3-Å resolution), the empty P116 (4-Å resolution) and the refilled P116 (3.5-Å resolution) have been deposited in the EMDB under the accession codes EMD-15274, EMD-15275 and EMD-15276, respectively. Model coordinates of original and empty P116 have been deposited in the PDB under the accession codes 8A9A and 8A9B, respectively. The mass spectrometry proteomics data have been deposited to the ProteomeXchange Consortium (http://proteomecentral.proteomexchange.org) via the PRIDE partner repository[42] with the dataset identifier PXD037758. Source data are provided with this paper.

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

## Acknowledgements

We thank M. R. Vabulas for valuable discussions and comments. We also thank the Frankfurt Center for Electron Microscopy for measurement time. We thank the Central Electron Microscopy Facility at the MPI of Biophysics in Frankfurt, which enabled us to collect the empty P116 dataset, particularly S. Welsch who assisted during the data collection. We thank L. Company and I. Fernández-Vidal for their support during MALS and mass spectroscopy measurements, A. Iborra (Servei de Cultius Cellulars, Anticossos Citometria, UAB) for his assistance with immunizing mice, D. Santos for his assistance in the radioactivity experiment and R. Pérez-Luque and D. Aparicio for their constant support and discussions. J. P. was funded by grants BIO2017-84166-R and PID2021-125632OB-C22 from the ministerio de Ciencia, Innovación y Universidades (MICINN, Spain). I. F. was funded by MICINN-Spain grant PID2021-125632OB-C21. A. S. F. was supported by the Deutsche Forschungsgemeinschaft (FR 1653/14-1 for MS and, FR 1653/6-3 for LS) and the Research Training Group iMOL (GRK 2566/1 for SM).

## Author contributions

L. S.: Planned and carried out the single-particle analysis; solved the P116 structure in the original, emptied and refilled state; proposed the mechanism of cholesterol uptake on the basis of the structural data; solved the structure of P116 with HDL. D. V.: initiated the project; prepared of the P116 gene synthesis; cloned two constructs, 13–957 and 30–657; did expression and purification tests; adapted the best conditions for its stabilization; did expression and purification of mutant W681A and analysis by MALS. The order of L. S. and D. V. in the author list was decided randomly. Both authors are entitled to use their names first when citing this work. D. V., L. S., I. F., A. S. F.: model building of the original, emptied and refilled P116 protein. D. V. and J. M.: emptying protocol; on-site assistance in the experimental preparation of the HDL-P116 interaction and the uptake of radioactive cholesterol (free and esterified). J. J., N. R., J. C. E.-G.: assessed uptake of radioactive cholesterol (free and esterified). S. M.: planned and carried out the single-particle sample preparation for the empty and refilled samples and the sample mixed with HDL. J. P. and M. M.: obtained hybridomas and monoclonal antibodies against P116 protein; immunolocalization of P116 by epifluorescence microscopy, time-lapse microcinematography. M. P. S.: Advised on single-particle experiments and structure determination procedures. J. D. L. and J. M.-C.: Mass spectrometry analyses of all samples prepared for single-particle analysis; lipidomics and proteomics analyses. I. F.: Designed and supervised research. A. S. F.: Designed and supervised research. I. F. and A. S. F.: Wrote the manuscript, with contributions from all authors.

## Competing interests

The authors declare no competing interests.

## Additional information

**Extended data** is available for this paper at https://doi.org/10.1038/s41594-023-00922-y.

**Correspondence and requests for materials** should be addressed to Ignacio Fita or Achilleas S. Frangakis.

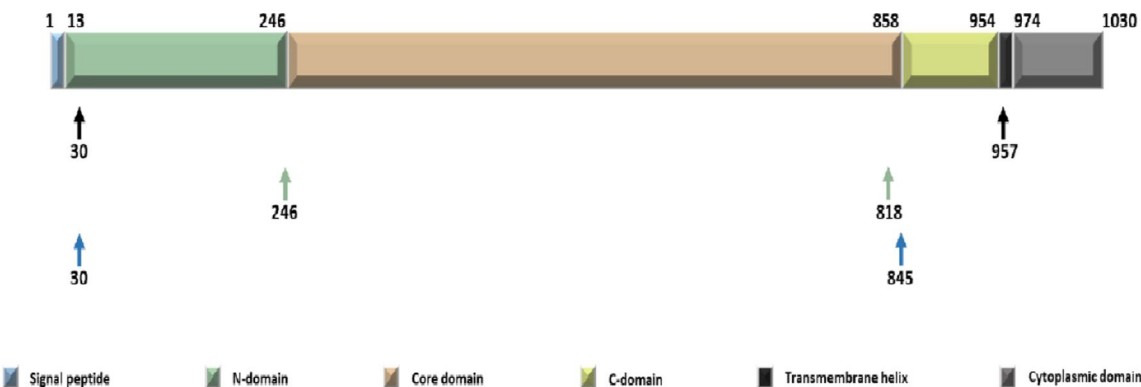

**Extended Data Fig. 1 | P116 constructs.** Overview of the different constructs (30–957, 30–818, 246–845) used for expression. For expression purposes a His-tag (KHHHHH) was added at the C-terminus. For the structural analysis by cryoEM the construct from 30-957 was used.

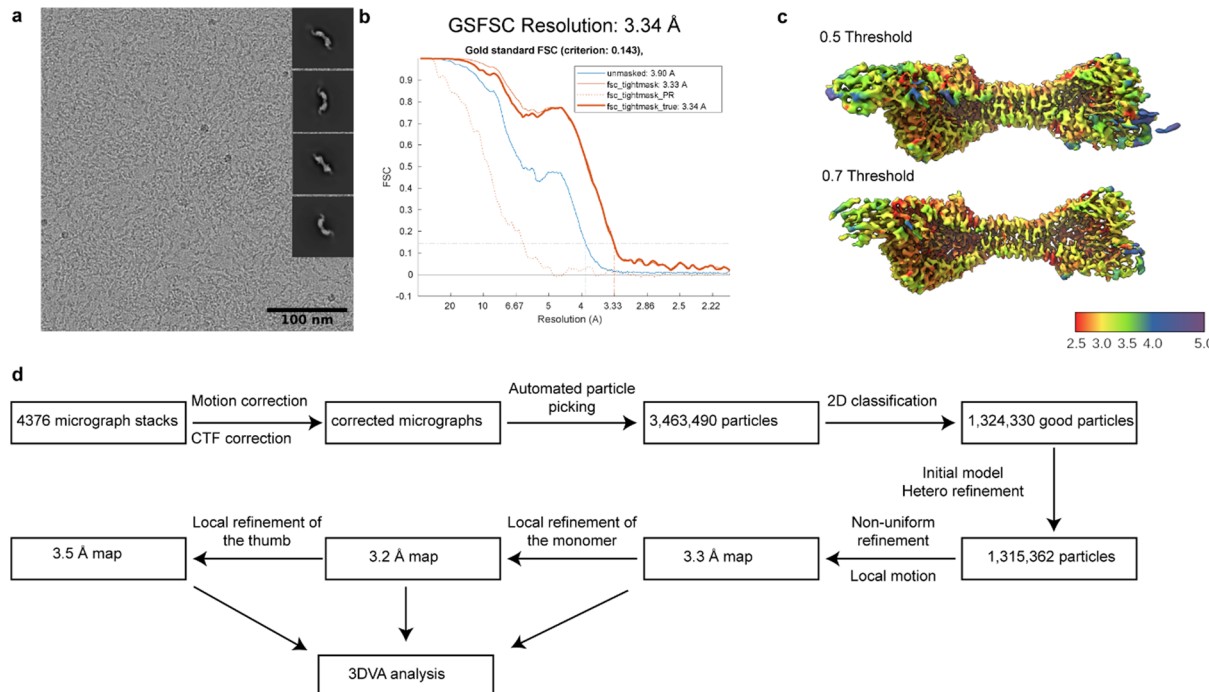

**Extended Data Fig. 2 | Overview of cryoEM processing of P116. (a)**
Representative Micrograph and obtained 2D classes used for template picking.
(**b**) Fourier shell correlation of P116 reports a final resolution of 3.3 Å according to
the 0.143 cut-off criteria. (**c**) Local resolution map at different thresholds of the
cryoEM density map (C1 symmetry) of P116 ranging from 2.5 to 5 Å. (**d**) Schematic
processing overview for the P116 construct. The processing for P116 empty, P116
refilled and P116 + HDL was carried out in a similar manner.

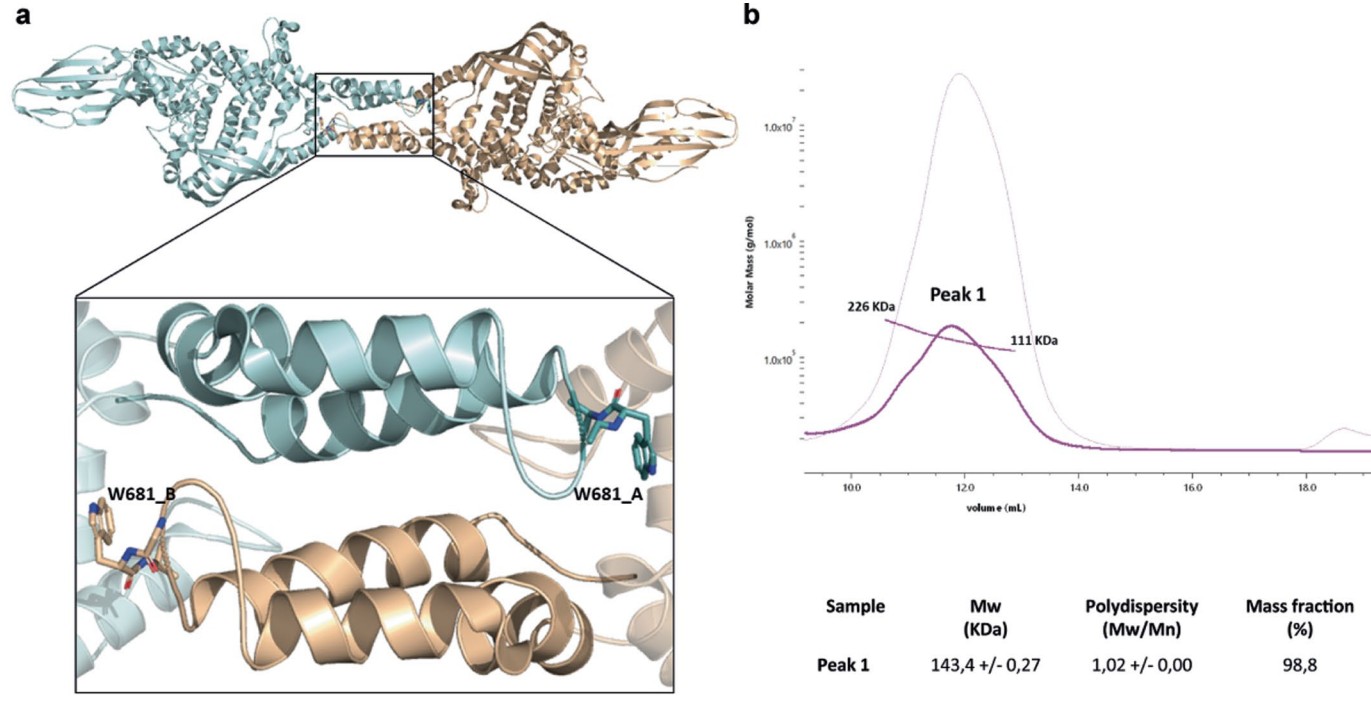

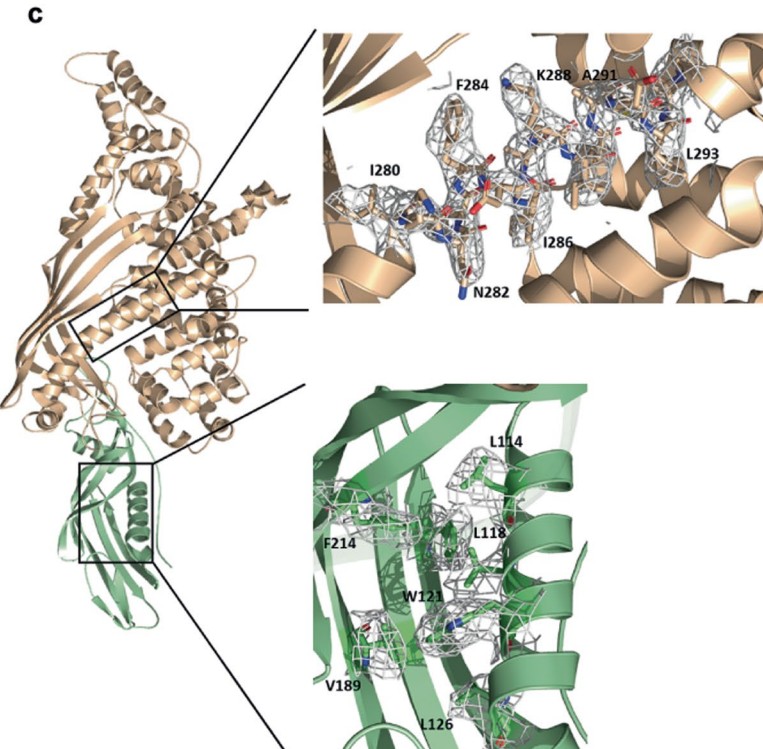

| Sample | Mw (KDa) | Polydispersity (Mw/Mn) | Mass fraction (%) |
|---|---|---|---|
| Peak 1 | 143,4 +/- 0,27 | 1,02 +/- 0,00 | 98,8 |

**Extended Data Fig. 3 | Dimerization interface and model building of P116.**
Detail of the dimerization zone with the two monomers and the TRP-681 residue of both chains contacting the opposite one. (**b**) SEC-MALS of P116 W681A. The greatest polydispersity of this sample is observed, which oscillates between a very large MW (molecular weight) range, and the clear decrease in size with respect to the WT due to the now predominance of the monomeric state. (**c**) Exemplary superimpositions of the cryoEM density and the PDB model for the core and N-terminal domain.

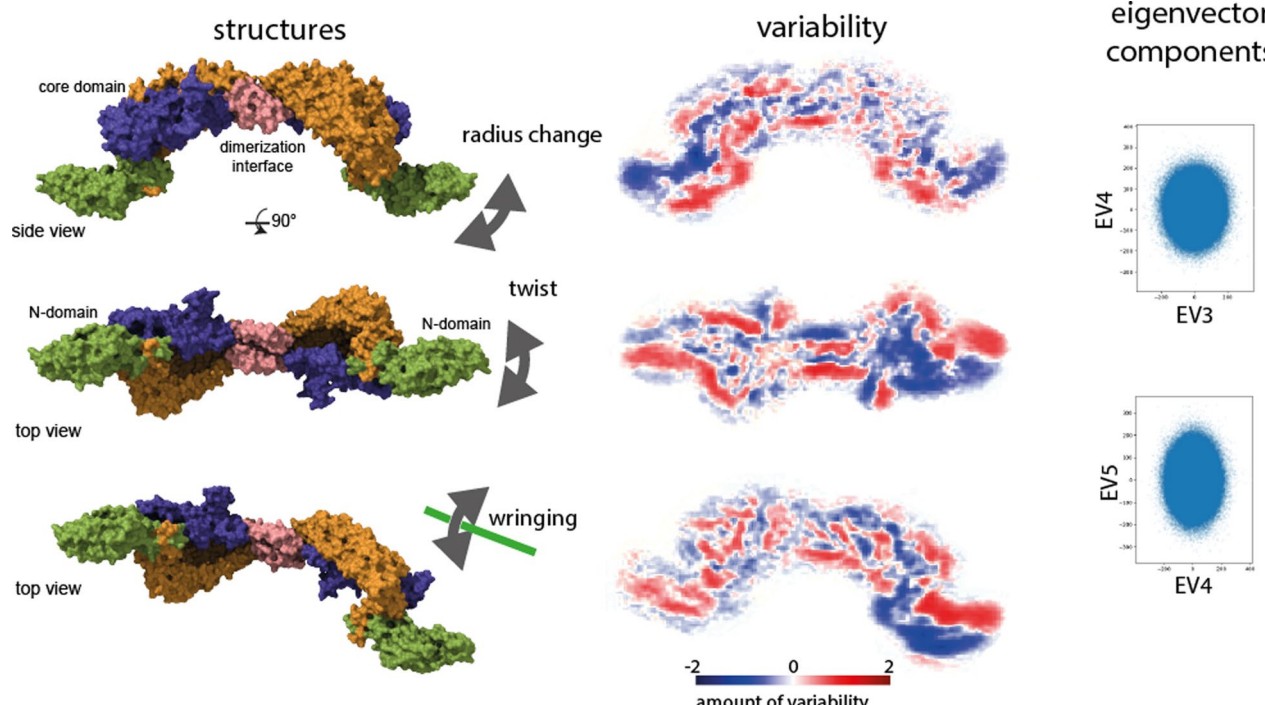

**Extended Data Fig. 4 | Flexibility of P116.** CryoEM classes display different flexibility modes. Individual reconstructions are shown on the left column. Their variability is shown in the middle column. Plotting of the individual eigenvector weights shows, that no clear distribution is visible and a smooth transition between the individual vibrational modes. Here 'eigenvector 3' (EV3) is plotted against 'eigenvector 4' (EV4) that display the largest resemblance to the radius change and 'eigenvector 4' (EV4) is plotted against 'eigenvector 5' (EV5) that display the largest resemblance to the twist.

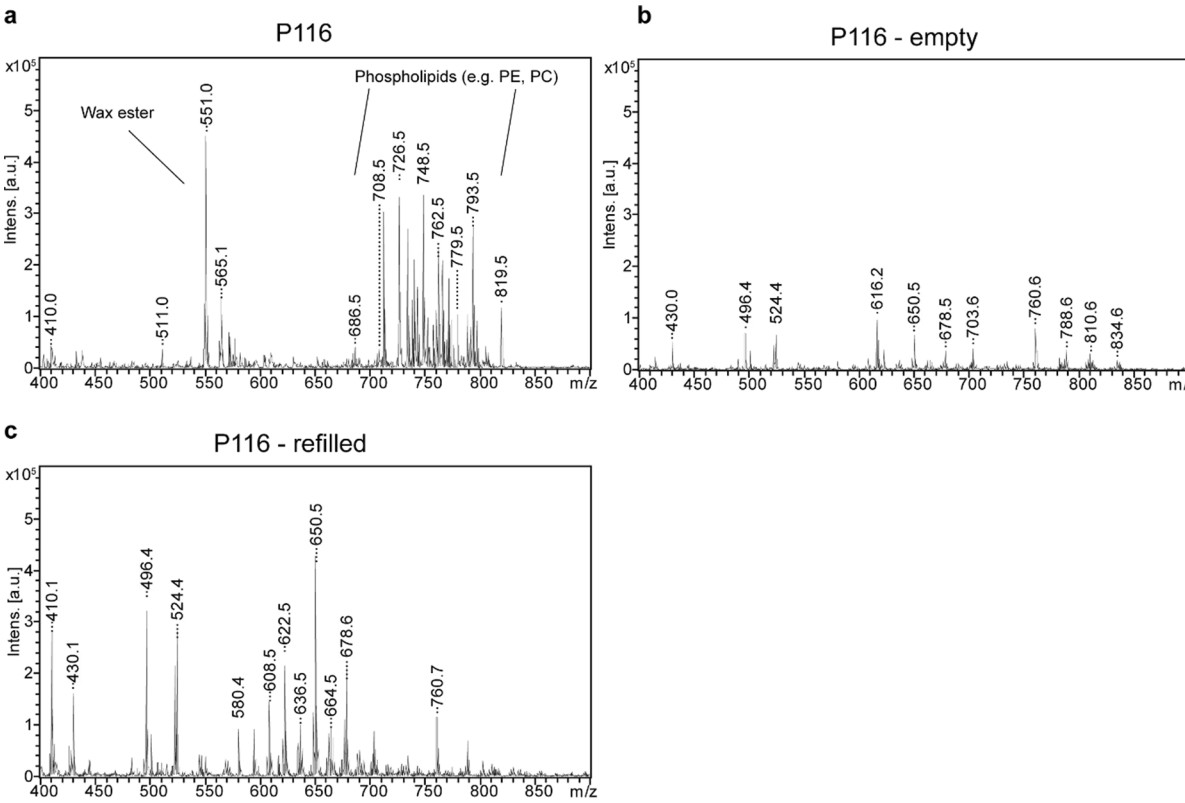

**Extended Data Fig. 5 | MALDI-TOF of P116, P116 empty and P116 refilled.** Individual low molecular weight MALDI-MS spectra of P116, P116-empty and P116-refilled as displayed in in Fig. 4d.

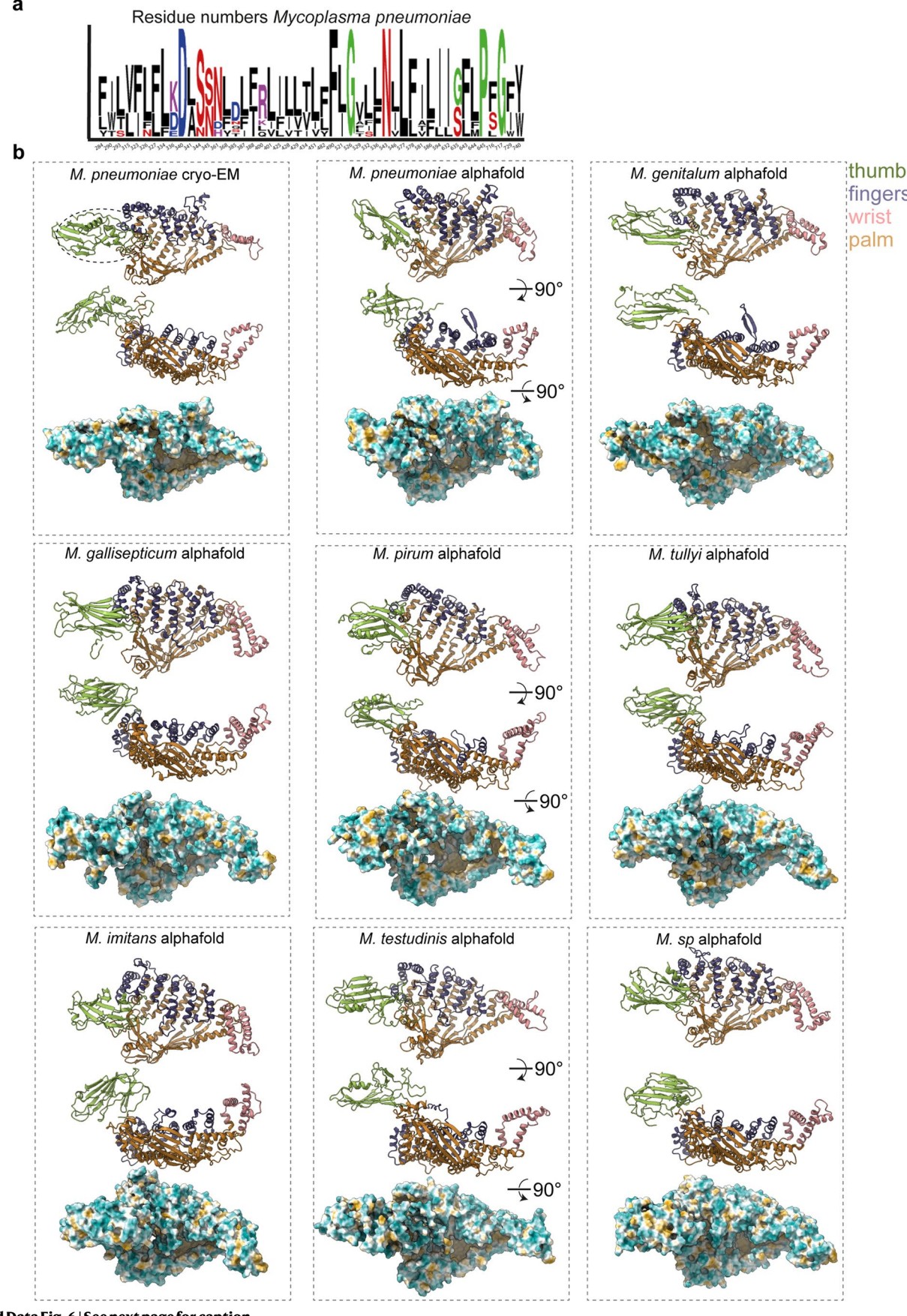

**Extended Data Fig. 6 | See next page for caption.**

**Extended Data Fig. 6 | P116 in other *Mycoplasma* species.** (**a**) Conserved (minimum 55%) amino acids pointing inwards the hydrophobic cavity. Black indicates amino acids with a hydrophobic character. Red indicates amino acids with a hydrophilic character. Blue indicates amino acids with a negative side chain. Purple indicates amino acids with a positive side chain. Green indicates other amino acids. (**b**) Predictions of P116 from different *Mycoplasma spp.* in comparison with the structure resolved with cryoEM. The N-terminal and the core domain have been predicted independently and subsequently fitted in the cryoEM density map of P116. The coloring of the models is identical to the Figs. 1–4.

**a  P116 empty**

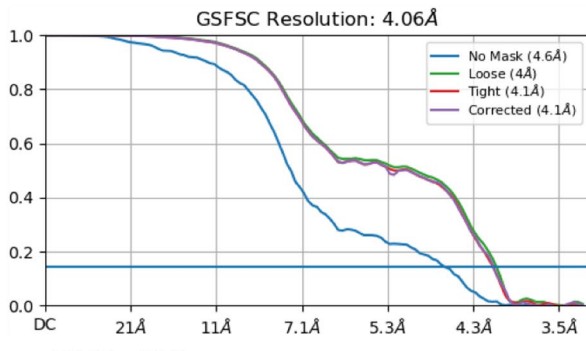

**b  P116 refilled**

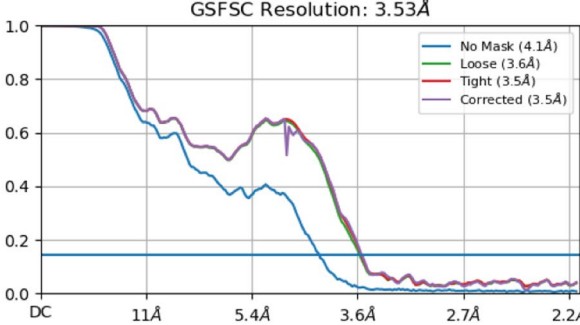

**c  P116 + HDL**

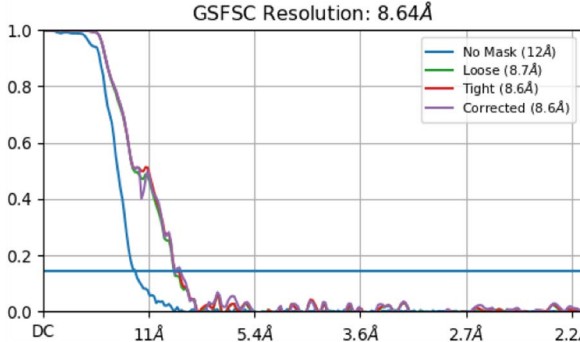

**Extended Data Fig. 7 | Fourier shell correlations of P116 empty, P116 refilled and P116 + HDL. (a)** Final reported resolution of P116 empty at 4 Å according to the 0.143 cut-off criteria. **(b)** Final reported resolution of P116 refilled at 3.5 Å according to the 0.143 cut-off criteria. **(c)** Final reported resolution of P116 + HDL at 9 Å according to the 0.143 cut-off criteria.

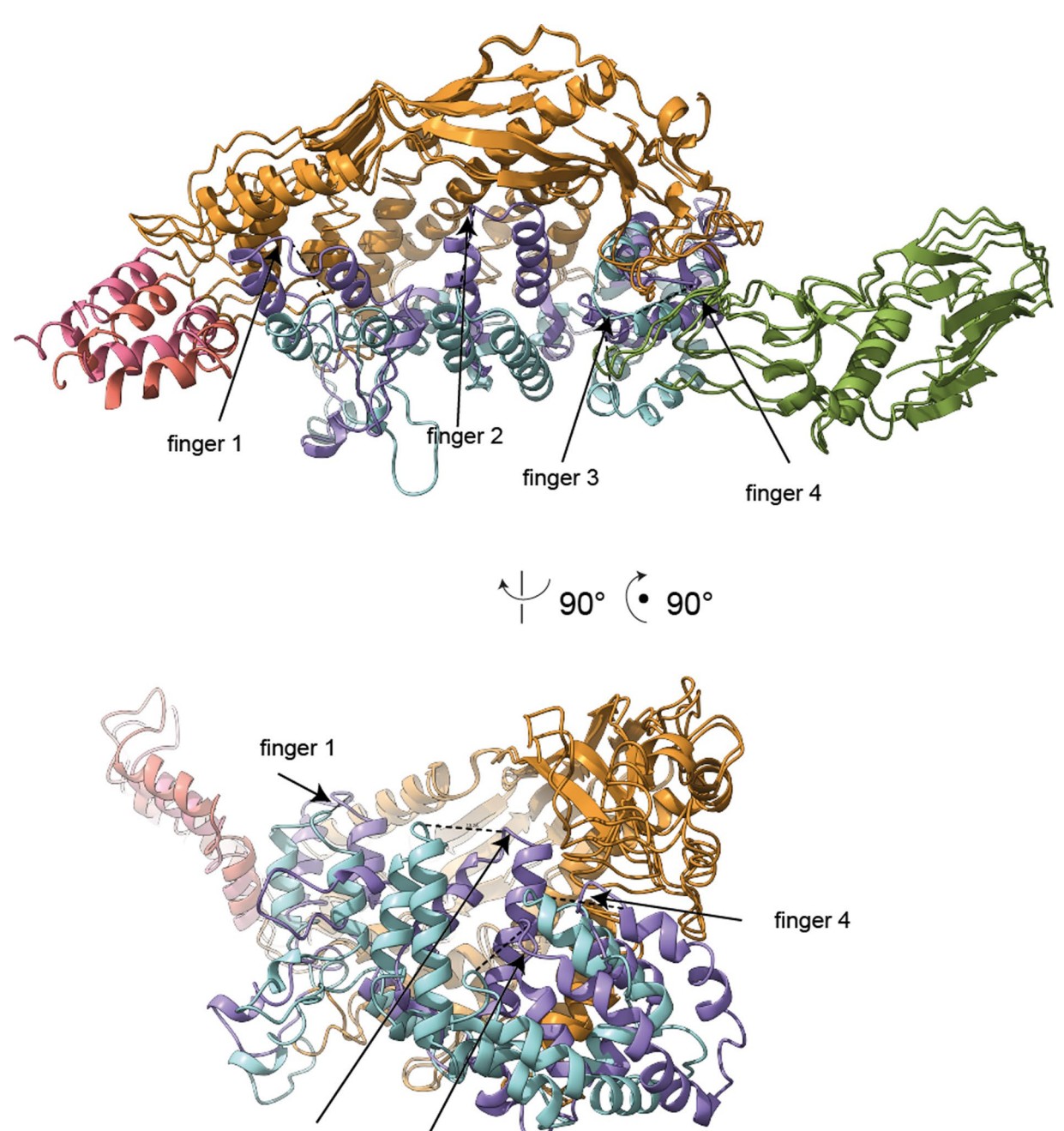

**Extended Data Fig. 8 | Conformational change between P116 and P116 empty.** Superimposition of one subunit of P116 with one subunit of P116 empty. The movement of the fingers 1-4 is indicated by dashed lines. Side view of the ribbon representation of the empty and full P116 shows that the fingers (in purple) have come closer to the core domain, massively reducing the available volume. Their new position is markedly different compared to the full P116 (shown in light blue). Finger 1 moved 8 Å sideways and towards the core, finger 2 has moved 13 Å towards the core and Finger 3 has moved 12 Å towards the core.

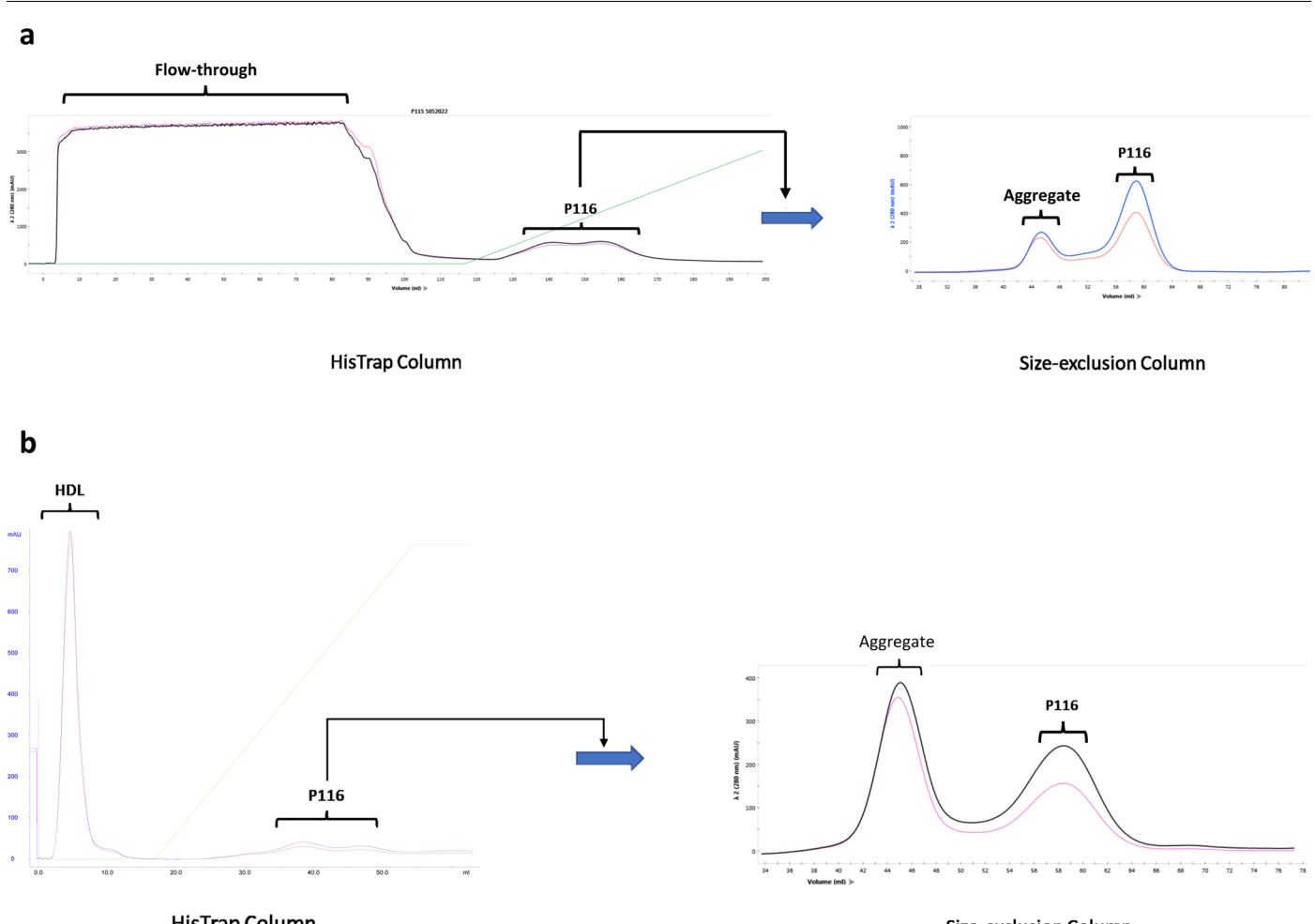

**Extended Data Fig. 9 | HisTraps and Size-exclusion running profiles of the purification.** (**a**) Running profile of the purification of the original P116 construct. (**b**) Running profile of the separation between P116 and HDL. The green slope indicates the increase of the imidazole concentration during the purification.

**a**

Mycoplasma pneumoniae tomogram

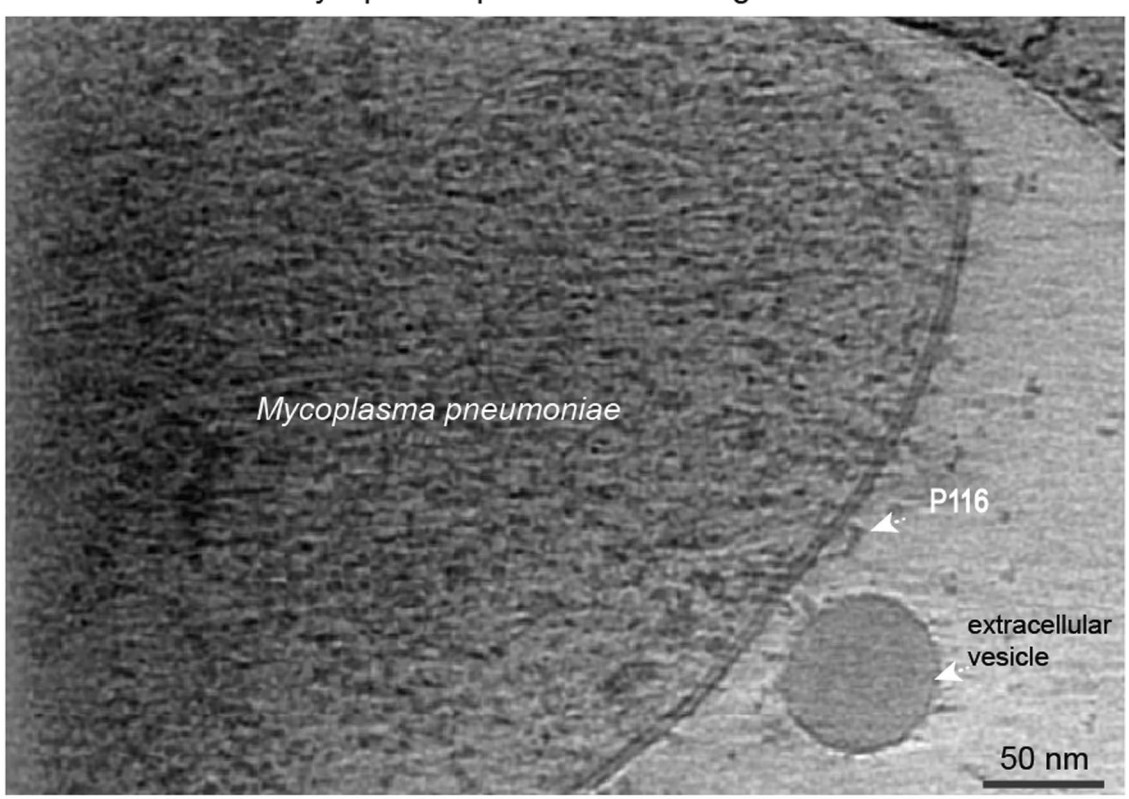

**b**

individual P116

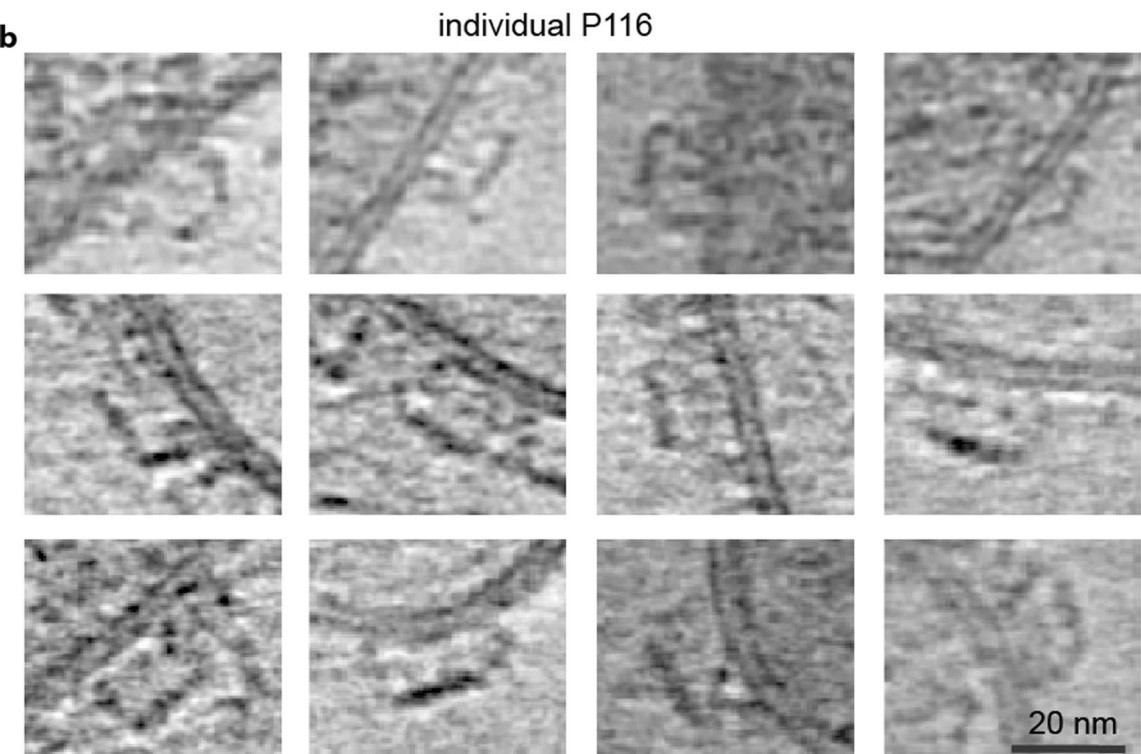

**Extended Data Fig. 10 | Cryo-electron tomogram of a *M. pneumoniae* cell.**
(**a**) Representative 10 nm thick slice of a tomographic reconstruction of an individual *M. pneumoniae* cell depicting a possible P116 protein on the surface of the *M. pneumoniae* membrane (One slice from a tomogram is shown – 10 tomograms were recorded) (**b**) Gallery of different possible P116 proteins from several tomographic reconstructions depicting P116 attached to the membrane (P116 molecules could be observed from three independent preparations). Tomograms were acquired with a defocus of −3 μm.

# Reporting Summary

## Statistics

For all statistical analyses, confirm that the following items are present in the figure legend, table legend, main text, or Methods section.

| n/a | Confirmed | |
|---|---|---|
| ☐ | ☒ | The exact sample size (*n*) for each experimental group/condition, given as a discrete number and unit of measurement |
| ☐ | ☒ | A statement on whether measurements were taken from distinct samples or whether the same sample was measured repeatedly |
| ☒ | ☐ | The statistical test(s) used AND whether they are one- or two-sided<br>*Only common tests should be described solely by name; describe more complex techniques in the Methods section.* |
| ☒ | ☐ | A description of all covariates tested |
| ☒ | ☐ | A description of any assumptions or corrections, such as tests of normality and adjustment for multiple comparisons |
| ☒ | ☐ | A full description of the statistical parameters including central tendency (e.g. means) or other basic estimates (e.g. regression coefficient) AND variation (e.g. standard deviation) or associated estimates of uncertainty (e.g. confidence intervals) |
| ☒ | ☐ | For null hypothesis testing, the test statistic (e.g. *F*, *t*, *r*) with confidence intervals, effect sizes, degrees of freedom and *P* value noted<br>*Give P values as exact values whenever suitable.* |
| ☒ | ☐ | For Bayesian analysis, information on the choice of priors and Markov chain Monte Carlo settings |
| ☒ | ☐ | For hierarchical and complex designs, identification of the appropriate level for tests and full reporting of outcomes |
| ☒ | ☐ | Estimates of effect sizes (e.g. Cohen's *d*, Pearson's *r*), indicating how they were calculated |

*Our web collection on statistics for biologists contains articles on many of the points above.*

## Software and code

Policy information about availability of computer code

| Data collection | Freeware and commercial code has been used as referenced in the manuscript.<br>Electron microscopy: Serial EM v3.8, EPU 2.12<br>Light microscopy: ORCA-Fusion<br>Mass spectrometry: Compass 2.0 |
|---|---|
| Data analysis | Freeware and commercial code has been used as referenced in the manuscript.<br>cryoSPARC v3.2.2, Artiatomi v0.1<br>LM: ImageJ<br>Mass spectrometry: MS dial lipidomics pipeline v 4.9, mTIC, R using the lipidr package, Frag-Pipe 18 using msfragger 3.5 |

For manuscripts utilizing custom algorithms or software that are central to the research but not yet described in published literature, software must be made available to editors and reviewers. We strongly encourage code deposition in a community repository (e.g. GitHub). See the Nature Portfolio guidelines for submitting code & software for further information.

## Data

Policy information about [availability of data](availability of data)

All manuscripts must include a [data availability statement](data availability statement). This statement should provide the following information, where applicable:
- Accession codes, unique identifiers, or web links for publicly available datasets
- A description of any restrictions on data availability
- For clinical datasets or third party data, please ensure that the statement adheres to our [policy](policy)

Cryo-electron microscopy densities of the original P116 density map (3.3 Å), the empty P116 (4 Å) and the refilled P116 (3.5 Å) have been deposited in the EM Data Base under the accession codes EMD-15274, EMD-15275 and EMD-15276, respectively. Model coordinates of original and empty P116 have been deposited in the PDB under the accession codes 8A9A and 8A9B, respectively. The mass spectrometry proteomics data have been deposited to the ProteomeXchange Consortium (http://proteomecentral.proteomexchange.org) via the PRIDE partner repository (Perez-Riverol et al, 2019) with the dataset identifier PXD037758.
Uniprot (P75556 Y213_MYCPN)

## Human research participants

Policy information about [studies involving human research participants and Sex and Gender in Research.](studies involving human research participants and Sex and Gender in Research.)

| | |
|---|---|
| Reporting on sex and gender | out studies did not involve human research participants |
| Population characteristics | n/a |
| Recruitment | n/a |
| Ethics oversight | n/a |

Note that full information on the approval of the study protocol must also be provided in the manuscript.

# Field-specific reporting

Please select the one below that is the best fit for your research. If you are not sure, read the appropriate sections before making your selection.

☒ Life sciences    ☐ Behavioural & social sciences    ☐ Ecological, evolutionary & environmental sciences

For a reference copy of the document with all sections, see [nature.com/documents/nr-reporting-summary-flat.pdf](nature.com/documents/nr-reporting-summary-flat.pdf)

# Life sciences study design

All studies must disclose on these points even when the disclosure is negative.

| | |
|---|---|
| Sample size | cryoEM:SupplementaryTable I. Sample size was chosen to such an amount in order to reach <4A resolution. Total particle size:<br>1.3 Mio particles for the purified P116<br>1.1 Mio particles for the empty P116<br>1.3 Mio particles for the refilled P116<br>58000 particle for the complex P116&HDL<br>Unsupervised classification was used to reject outliers.<br>LM:  Five separate experiments for testing the antibodies and 3 experiments for testing the controls.<br>Counts of motile and non-motile mycoplasma cells were performed before adding antibodies. Once antibodies were added, the number of cells was determined after 10 min of incubation (table attached). The mean and standard deviation were computed from cell counts and no further statistical analyses were performed. |
| Data exclusions | cryoEM: All particles were included, classification was used to cluster the data set.<br>LM: No data was excluded<br>Mass-Spec: No data was excluded |
| Replication | cryoEM: Refinements and averages were done with random individual half sets. All experimental findings could be replicated.<br>LM: Five independent microcinematography replicates were made for each antibody and three replicates were performed for control experiments with no antibodies. All findings could be replicated.<br>3H Cholesterol Experiment: This experiment was performed once (n=1) as a proof of concept experiment to initiate the thorough analyses by LC-MS/MS.<br>LC-MS/MS: Proteomics and Lipidomics experiment was conducted as three independent replicates. All findings could be replicated. |
| Randomization | cryoEM: Single particle analysis and classification was done with CryoSparc. Refinements and averages were done with random individual half sets. |

| Blinding | LM: Field of view was randomly selected, all single cells were included in the analysis.<br>Mass-Spec: n/a |
| | cryoEM: Investigators were not blinded during grouping since it was computationally performed. HDL cholesterol transfer rate: Technician was blinded regarding the source of tritium-labeled HDL and P116 form.<br>LM: Investigators were not blinded.<br>Mass-Spec: n/a |

# Reporting for specific materials, systems and methods

We require information from authors about some types of materials, experimental systems and methods used in many studies. Here, indicate whether each material, system or method listed is relevant to your study. If you are not sure if a list item applies to your research, read the appropriate section before selecting a response.

## Materials & experimental systems

| n/a | Involved in the study |
|---|---|
| ☐ | ☒ Antibodies |
| ☒ | ☐ Eukaryotic cell lines |
| ☒ | ☐ Palaeontology and archaeology |
| ☐ | ☒ Animals and other organisms |
| ☒ | ☐ Clinical data |
| ☒ | ☐ Dual use research of concern |

## Methods

| n/a | Involved in the study |
|---|---|
| ☒ | ☐ ChIP-seq |
| ☒ | ☐ Flow cytometry |
| ☒ | ☐ MRI-based neuroimaging |

## Antibodies

| Antibodies used | LM: Antibodies were generated within this work as described in the material and methods section.<br>Goat anti-mouse Alexa 555 (Invitrogen, Waltham, USA) secondary antibody: dilution: 1/2000;<br>P116 polyclonal antibodies: dilution 1/2000;<br>P1 polyclonal antibodies: dilution 1/2000;<br>Monoclonal antibodies: dilution 1/10: validation: indirect ELISA, western blot and immunofluorescence staining;<br><br>3H Experiment: A commercial anti-apoA-I-based assay is described in the material and methods section. |
| --- | --- |
| Validation | Supernatants from hybridoma cell lines derived from single fused cells were first investigated by indirect ELISA screening against the recombinant P116 ectodomain. Positive clones were also tested by Western blot against protein profiles from M. pneumoniae cell lysates and by immunofluorescence using whole, non-permeabilized M. pneumoniae cells (see below). Only those clones with supernatants revealing a single 116 kDa band in protein profiles and also exhibiting a consistent fluorescent staining of M. pneumoniae cells were selected and used in this work. ApoA1 was determined by an immunoturbidimetric assay, using a commercial kit, including standards for calibration and internal controls, adapted for a COBAS 6000 autoanalyzer (Roche Diagnostics, Rotkreuz, Switzerland).<br><br>LM: validation for Alexa 555: https://www.thermofisher.com/order/genome-database/dataSheetPdf?producttype=antibody&productsubtype=antibody_secondary&productId=A-21424&version=256; company names: Invitrogen; Cat Number: A21424.<br>P116 polyclonal antibodies: dilution 1/2000; validation: indirect ELISA, and Western blot; obtained in-house.<br>P1 polyclonal antibodies: dilution 1/2000; validation: indirect ELISA, and Western blot; obtained in-house.<br>Monoclonal antibodies: dilution 1/10: validation: indirect ELISA, western blot and immunofluorescence staining; obtained in-house.<br>Further validation is provided in the M&M section |

## Animals and other research organisms

Policy information about studies involving animals; ARRIVE guidelines recommended for reporting animal research, and Sex and Gender in Research

| Laboratory animals | Mice:<br>- Two BALB/c females eight-week old were immunized to obtain polyclonal antibodies and monoclonal antibodies against P116.<br>- Two BALB/c females eight-week old were immunized to obtain polyclonal antibodies against P1.<br>The experimental procedures to immunize mice and obtaining monoclonal antibodies were approved by the Ethics Committee on Animal and Human Experimentation from the Universitat Autònoma de Barcelona by the document CEEAH 1002R3R2R. |
| --- | --- |
| Wild animals | No wild animals were used in the study. |
| Reporting on sex | Two BALB/c females eight-week old were used |
| Field-collected samples | no field collected samples were used in the study |

Ethics oversight | The experimental procedures to immunize mice and obtaining monoclonal antibodies were approved by the Ethics Committee on Animal and Human Experimentation from the Universitat Autònoma de Barcelona by the document CEEAH 1002R3R2R. Also provided in the manuscipt in M&M section / timelapse

Note that full information on the approval of the study protocol must also be provided in the manuscript.

