## [Peer Review File · Nature Structural & Molecular Biology]

Peer Review Information

Manuscript Title: Essential protein P116 extracts cholesterol and other indispensable lipids for Mycoplasmas

Corresponding author name(s): Achilleas S. Frangakis, Ignacio Fita

Reviewer Comments & Decisions:

Decision Letter, initial version:
--

Message: 7th Oct 2022

Dear Professor Frangakis,

Thank you again for submitting your manuscript "Immunodominant protein P116 from *M. pneumoniae* transports cholesterol and essential lipids". We now have comments (below) from the 3 reviewers who evaluated your paper. In light of those reports, we remain interested in your study and would like to see your response to the comments of the referees, in the form of a revised manuscript. Please be sure to address/respond to all concerns of the referees in full in a point-by-point response and highlight all changes in the revised manuscript text file.

We appreciate the requested revisions are extensive. We thus expect to see your revised manuscript within 6 months. If you cannot send it within this time, please let us know. We will be happy to consider your revision as long as nothing similar has been accepted for publication at NSMB or published elsewhere. Should your manuscript be substantially delayed without notifying us in advance and your article is eventually published, the received date would be that of the revised, not the original, version.

Reporting Summary:

When submitting the revised version of your manuscript, please pay close attention to our [Digital Image Integrity Guidelines](https://www.nature.com/nature-portfolio/editorial-policies/image-integrity).

We require deposition of coordinates (and, in the case of crystal structures, structure factors) into the Protein Data Bank with the designation of immediate release upon publication (HPUB). Electron microscopy-derived density maps and coordinate data must be deposited in EMDB and released upon publication. Deposition and immediate release of NMR chemical shift assignments are highly encouraged. Deposition of deep sequencing and microarray data is mandatory, and the datasets must be released prior to or upon publication. To avoid delays in publication, dataset accession numbers must be supplied with the final accepted manuscript and appropriate release dates must be indicated at the galley proof stage. Please find the complete NRG policies on data availability at <http://www.nature.com/authors/policies/availability.html>.

[Redacted]

Kind regards,
Florian

Dr Florian Ullrich
Associate Editor, Nature
Consulting Editor, Nature Structural & Molecular Biology
ORCID 0000-0002-1153-2040

Referee expertise:

Referee #1: M. tuberculosis biology

Referee #2: cryo-EM

Referee #3: cryo-EM

Reviewers' Comments:

Reviewer #1:

Remarks to the Author:

The manuscript entitled "Immunodominant protein P116 from *M. pneumoniae* transports cholesterol and essential lipids" reports a cryoEM structure of the essential surface protein, P116 establish that this protein binds various lipids.

Overall this is a well written and concise body of work that should inform in the area of lipid binding/transporting proteins.

M. pneumoniae is a cell wall deficient bacterium with a minimal genome which requires numerous host-derived nutrients including lipids to grow and replicate. Briefly, this study determined cryoEM structures of the homodimeric P116 ectodomain which revealed a novel structure containing a large hydrophobic cleft and various lipids co-purified with P116 isolated from *E. coli*. The authors next demonstrate that stripping P116 with a Triton X100 procedure the co-purified lipids in P116 could be removed and the cryoEM structure of the stripped protein had a smaller hydrophobic cleft and incubating the stripped protein with serum or serum lipoproteins "refilled" the cavity. The cryoEM structure of the "refilled" P116 resembled shape and size of the protein isolated from *E. coli*.

Using a radioactivity transfer assay the authors provide some evidence that P116 preferentially binds free cholesterol over esterified cholesterol upon HDL "refilling" of P116. Additionally, the mass spec-based analysis of lipids that occupied P116 following "refilling" with serum or lipoproteins were phosphatidyl choline's, sphingomyelin's, cholesterol, triacylglycerol, free fatty acids, and lysophosphatidic acids.

Major suggestions:

- Please change the title. Nowhere in this manuscript has the authors demonstrated cholesterol or lipid transport. Perhaps emphasize binding of lipids by the essential protein, P116?
- Fig 4B, Fig S6 and line 146 indicate that sphingomyelin co-purifies with P116 from E. coli. As far as I know, E. coli does not produce sphingomyelin. Please explain.
- Please provide evidence that the P116 prep analyzed in lipidomic experiments did not have any serum lipoprotein carryover. Any carryover of lipoproteins would confound these analyses and the authors indicate that P116 and HDL do bind, Fig 4F.
- Please provide evidence that the P116 prep analyzed in 3H-transfer experiments did not have any HDL carryover as indicated in lines 211-213. Any carryover of HDL would confound these data and the authors indicate that P116 and HDL do bind, Fig 4F.
- Please indicate what lipids/detergent are present in P116 following the TritonX 100 stripping in Fig 4C.
- It was not clear to this reviewer where the lipidomic data for cholesterol is in the supplement positive ion mode Excel sheet. It looks like the data is for cholesterol-ester and it is labeled cholesterol in Fig 4C. This is important because it could contradict the findings in the 3H-tracer study.

Minor Suggestions:

- Please provide an abbreviation list for the lipid species in Fig 4C and 4D.
- Might be useful to more clearly indicate cholesterol ester in the heatmaps of Fig 4 following the refilling with HDL and serum to support the 3H-tracing study.
- The authors should revisit the literature around how other bacteria transport lipids and cholesterol, lines 62-63. For example (PMID: 18334639, PMID: 28708968, PMID: 18955493)

Reviewer #2:

Remarks to the Author:

The Sprankel et al manuscript provides a structural and functional analysis of the previously uncharacterized protein P116 from *Mycoplasma pneumoniae*. The protein in study is essential for *M. pneumoniae*, previously known to potentially contributing to

adhesion to host cells. The authors used cryo-electron microscopy to solve the 3D structures of several purified proteins with and without lipids, with resolution higher than 4 Å. They described the flexibility of a homodimer and performed experiments to prove its binding of lipids. They also demonstrated that P116 has a new, unique fold with a totally hydrophobic cavity in the middle of each monomer. Authors suggested that they can describe the mechanism by which Mycoplasmas can extract lipids from the environment. As such, the study represents a contribution.

The manuscript is suitable for publication in Nature Structural and Molecular Biology after major revision. The points are listed below:

- 1) From the introduction it is unclear why the authors choose this protein for this study.
- 2) In Figure 1a - the image of fluorescently labeled cells is not sufficiently clear. To better demonstrate in immunofluorescence analyses that proteins are located on the cell surface, it is necessary to use a confocal microscope and additionally stain the plasma membrane with specific dyes, and compare the signal from the membrane and protein. Otherwise from Figure 1a it is not obvious that P116 is located ON the surface and not inside the cell.
- 3) Also, in the legend "Structure of P116 and its localization in Mycoplasma pneumoniae cells" - Consider swapping 'localization' and 'structure'.
- 4) Line 106-107 "...flexibility with many vibrational modes, as classification illustrates (Supplementary Figure 5) - but there are not many classes on Supplementary Figure 5, just two. Besides, 3D classification is omitted in the Supplement figures. It should be present.
- 5) Figure 3d does not differ from Supplementary Figure 5 in color, shape, and orientation. Assuming that in Supplementary Figure 5 a filled P116 is presented, how can we understand that Figure 3d reflects the refilled P116?
- 6) Supplemental Figure 2b - why does the corrected curve on the FSC plot have such a 'spike' in the middle (purple)? Could it be a result of overfitting?
- 7) Supplemental Figure 2c - there is no explanation for why there are two maps present (different threshold?) and why the dimers are not symmetrical. Does it reflect C1 symmetry?
- 8) Line 192 and below: "P116 is conformationally flexible" - it has already been mentioned several times before that it is flexible, why repeat this again? The whole paragraph can be easily omitted.
- 9) Line 250 "Of ~46,000 particles that were identified as HDL, ~25,000 were attached to P116" - this is also not visible on Figure 4f. Classes are too small. Consider adding arrows to point to the extra-density, which belongs to HDL. Also, the gallery of free HDL particles is necessary.
- 10) Line 524-526 "Due to the flexibility of P116 and the variability of HDL, only one subunit of P116 can be seen at this threshold. Reducing the threshold causes the second subunit to appear." - show this in the supplement.
- 11) On a tomogram in Supplementary Figure 10 only one curved density is visible; why it reflects an arrangement of P116 with respect to the Mycoplasma membrane is not clear. Since Mycoplasma is a tiny cell, it will be useful, instead of one tomogram, to collect more cryo-images of cell P116 proteins on the surface, and, again, demonstrate via a gallery.
- 12) In the Discussion section, Authors claim that they suggest a mechanism for extracting the lipids from the environment. Here, the schematic will be useful to illustrate their conclusions.

Reviewer #3:

Remarks to the Author:

In this manuscript, Sprankel et al. describe the structure of the immunodomain protein P116 and its mode of operation. This protein is essential for the life cycle of the pathogen *Mycoplasma pneumoniae*. The authors find how the bacteria takes lipids from the host cell into its own. Surprisingly, P116 assembles as a dimer with two hydrophobic cavities, exhibiting a new protein fold. The authors determined the structure of the protein in both filled and empty state. Most impressively, the protein folded into a novel fold that has never been reported. The authors show that the protein can directly extract lipids from vesicle.

Therefore, this work contains impressive novel information that discover a protein machinery that can extract membranes and cholesterol from its environment. Besides exciting microbiology and protein structure this work may be of interest to biotechnology since a protein that can modify the content of membrane in a cell. I fully convinced that the work is spectacular and should be accepted for publication in NSMB.

1. It is not clear how the protein releases the lipids into the membrane of the bacteria and how does it identify the host from its own. Is it flexibly attached to the membrane of the bacteria ? How does it distinguish between the bacterial membrane and the host membrane ?
2. Please clarify what do you meant by arbitrary view (Fig. 3b)

Author Rebuttal to Initial comments

We would like to thank all reviewers for their very constructive comments and suggestions. Our responses are in green font. Manuscript changes are *italicized*.

Reviewers' Comments:

Reviewer #1:

Major suggestions:

- Please change the title. Nowhere in this manuscript has the authors demonstrated cholesterol or lipid transport. Perhaps emphasize binding of lipids by the essential protein, P116?

We thank the reviewer for this suggestion, and we changed the title according to their suggestion to:

“Essential protein P116 extracts cholesterol and other indispensable lipids for Mycoplasmas”

in order to highlight the rather broad spectrum of lipid binding shown for P116.

- Fig 4B, Fig S6 and line 146 indicate that sphingomyelin co-purifies with P116 from *E. coli*. As far as I know, *E. coli* does not produce sphingomyelin. Please explain.

We thank the reviewers for pointing this out. To clarify: No significant amounts of PC or SM were in the initial p116 sample; we only detect traces which could well derive from the yeast-based medium the *E. coli* grew in for p116 production. Only after refilling is an enrichment for PC and SM seen. Predominant phospholipid classes in the original p116 sample are phosphatidylethanolamine (PE) and phosphatidylglycerol (PG) which are present in *E. coli*. We have modified the respective text sections to reflect this better. The text now reads:

“Mass spectrometry analysis of the same samples from which the structure of P116 was determined (see Materials and Methods) showed the presence of several lipid species, predominantly phosphatidylethanolamine (PE) and phosphatidylglycerol (PG) lipids, as well as wax esters (Figure 4b-d and Supplementary Figure 6).”

- Please provide evidence that the P116 prep analyzed in lipidomic experiments did not have any serum lipoprotein carryover. Any carryover of lipoproteins would confound these analyses and the authors indicate that P116 and HDL do bind, Fig 4F.

From our proteomics data we see that we do have a small carryover of HDL-associated apolipoproteins, mainly APOAI, in the refilled sample – although in a very low proportion compared to p116.

We discuss this point now more clearly in the manuscript. We show that the relative and specific enrichment of certain lipids and the distribution of the lipids in the serum is significantly different compared to refilled P116 (i.e. the lipid profile changes significantly between serum/refilled p116). If there was an unspecific uptake or the lipid distribution was the result of lipoprotein carryover, the distribution would be similar to that of serum, which is not the case.

The text now reads (a new paragraph):

“To analyze lipoprotein carryover in the FBS-refilled P116, we conducted an additional proteomics LC-MS/MS experiment (Supplementary Table V) using ultrasensitive, ion-mobility-assisted LC-MS/MS. Here we observed limited lipoprotein carryover into the refilled sample. However, based on peptide spectrum match (PSM) numbers and intensity values, we found P116 to be over 30-fold more abundant than the lipoproteins in the refilled sample. If the lipid spectrum in the FBS-refilled P116 sample originated from lipoprotein carryover, we would expect a similar distribution

of the lipid classes in both samples. In fact, we observed a specific enrichment of PC and SM in the FBS-refilled sample, while TG, the most abundant lipid class in the serum, was decreased and was barely detectable. Thus, while P116 can extract a large range of lipids, it shows a preference for selected lipid species (Figure 4c, 4d and Supplementary Table IV). We conclude that the lipid composition in the FBS-refilled P116 sample can be attributed predominantly to P116 itself and not to lipoprotein carryover. ”

We also provide an additional table (Supplementary table V) with the lipids and apolipoprotein IDs and a whole section in the Materials and Methods.

- Please provide evidence that the P116 prep analyzed in 3H-transfer experiments did not have any HDL carryover as indicated in lines 211-213. Any carryover of HDL would confound these data and the authors indicate that P116 and HDL do bind, Fig 4F.

The 3H-transfer experiment was specific for free and esterified cholesterol and it does not reach the levels of sensitivity as the lipidomics analysis, which we explained above. Nevertheless, we could not detect APOA1 in the sample, indicating that it is free of HDL. Importantly, the 3H-transfer experiment was performed as a proof of concept that initiated the thorough and conclusive lipidomics analysis.

We now include detailed information about the separation of P116 from HDL and about how the controls by immune detection of APOA1 were performed.

The text in the manuscript has been modified and the Material and Methods were extended for better clarification. The text in the manuscript now reads:

*“A significant fraction of the HDL-³H] radiotracer was detected in the P116 samples that had been incubated with HDL and then separated from it by purification (Material and Methods and **Supplementary Figure 10**), indicating a net transfer of both cholesterol and cholesterol ester between HDL and P116. The total absence of the most abundant HDL protein (APOA1), cross-checked by immune detection (Material and Methods and **Supplementary Table III**), verified that no HDL remnants had contaminated the purified P116 samples.”*

The text in the Materials and Methods now reads:

“HDL and P116 were separated by a HisTrap HP affinity and size exclusion columns (Supplementary Figure 10)....

“Total cholesterol levels in the HDL fraction were determined enzymatically by using a commercial kit adapted for a COBAS 6000 autoanalyzer (ref 03039773190, Roche Diagnostics, Rotkreuz, Switzerland). Human APOA1 levels were determined in both the HDL and purified P116 fractions by an assay (ref 3032566122, Roche Diagnostics) that used anti-APOA1 antibodies that react with the antigen in the sample to form antigen/antibody complexes which, after agglutination, were measured turbidimetrically in the COBAS 6000 autoanalyzer (Supplementary Table III).”

In addition to Table I (Relative transfer of (esterified) cholesterol from HDL to P116) we also provide in the supplementary material Supplemental Table III (below), which shows the relative level of ApoA1 in both HDL in the radioactive assay and P116 after incubating with HDL, demonstrating that the ApoA1 could not be detected in the latter.

Relative levels of ApoA1 in the P116 samples used in the radioactive assays

	ApoA1 (mg/mL)
HDL *	
Free cholesterol-radiolabeled form	0.2272
Esterified cholesterol-radiolabeled form	0.2272
Purified P116**	
Empty	0.00 (Undetected)
Refilled	0.00 (Undetected)

* Corresponding to P116 samples incubated with HDL

** Corresponding to P116 samples purified after the incubation with HDL

- Please indicate what lipids/detergent are present in P116 following the TritonX 100 stripping in Fig 4C.

We thank the reviewer for this comment. After the stripping with Triton X, stringent washing was applied to remove detergents prior to LC-MS analysis as these would heavily interfere with the measurements. For clarification and better overview, we added the main lipid class labels to figure

4C. All remaining lipid IDs and relative quantities are (already) included in **Supplementary table 4** (p116-empty), which is (already) provided as an Excel file.

Figure 4 was updated to comply with the reviewer's comment

- It was not clear to this reviewer where the lipidomic data for cholesterol is in the supplement positive ion mode Excel sheet. It looks like the data is for cholesterol-ester and it is labeled cholesterol in Fig 4C. This is important because it could contradict the findings in the 3H-tracer study.

We clarify this now to avoid misunderstandings: Cholesterol is labelled as "ST 27:1;O" according to the lipid shorthand notation in the table, but as a single compound not highlighted in the heatmap (fig 4C). The generalisation in the heatmap might be misleading in this regard. Cholesterol-esters are present as "CE" labelled compounds, other sterol esters are marked as "SE".

We have replaced the label in Figure 4e with "Sterol lipids", which is the general class to which cholesterol also belongs. In addition, we specifically point to the band that represents cholesterol, and we mention this now clearly in the text.

Minor Suggestions:

- Please provide an abbreviation list for the lipid species in Fig 4C and 4D.

We now provide in the figure legend the abbreviations for each main lipid class in the **Figure 4 C-D**

The lipid shorthand notation used is according to Liebisch et al ([https://www.jlr.org/article/S0022-2275\(20\)60017-7/fulltext](https://www.jlr.org/article/S0022-2275(20)60017-7/fulltext)) and follows the LipidMaps classification system

- Might be useful to more clearly indicate cholesterol ester in the heatmaps of Fig 4 following the refilling with HDL and serum to support the 3H-tracing study.

We have relabelled sterols in Figure 4 and mentioned that cholesterol is part of them in the figure legend.

- The authors should revisit the literature around how other bacteria transport lipids and cholesterol, lines 62-63. For example (PMID: 18334639, PMID: 28708968, PMID: 18955493)

We would like to thank the reviewer for their suggestion, and we have incorporated some references in the introduction.

The introduction now reads:

“Comprehensive studies on other cholesterol-utilizing bacteria are largely lacking; the best characterized organism of this group is Mycobacterium tuberculosis, for which it has been proposed that an ABC transporter homolog and other genes from the mce4 operon are involved in the cholesterol uptake (9). The utilization of cholesterol as a carbon source enables M. tuberculosis to manifest long-term infections (10). For M. pneumoniae it was shown that they survive long-term in cholesterol rich atherosclerotic plaques (11).”

Reviewer #2:

1) From the introduction it is unclear why the authors choose this protein for this study.

Originally P116 was thought to be an adhesion protein and it was described as the second most immunogenic protein after P1. As we characterised P1 we also wanted to characterize P116. Along the way we discovered that P116 had a different role.

We rewrote the introduction:

“

In this work, we report the structural and functional characterization of P116. This protein was originally reported to contribute to host-cell adhesion. Furthermore, P116 is an essential protein for the viability of *M. pneumoniae* cells and strongly immunogenic, thus making it a promising target for therapeutics.”

2) In Figure 1a - the image of fluorescently labeled cells is not sufficiently clear. To better demonstrate in immunofluorescence analyses that proteins are located on the cell surface, it is necessary to use a confocal microscope and additionally stain the plasma membrane with specific dyes, and compare the signal from the membrane and protein. Otherwise from Figure 1a it is not obvious that P116 is located ON the surface and not inside the cell.

We agree with the reviewer that the resolution can be improved. However, our major aim was to show that P116 is distributed all over the cell and not localized at the tip as P1. Following the reviewer's suggestion, we tried to improve the imaging with a confocal microscope, but the images did not improve. Given the tiny sizes of the Mycoplasma cells ~200nm we come very close to the diffraction limit, for any not super-resolution light microscopy technique.

Proof that P116 is indeed a surface protein was provided elsewhere using immune light microscopy and electron microscopy. This paper is cited as reference number 12.

We have changed the title of the section to:
"P116 is evenly distributed on the cell surface"

3) Also, in the legend "Structure of P116 and its localization in Mycoplasma pneumoniae cells" – Consider swapping 'localization' and 'structure'.

We thank the reviewer for this suggestion, which we implemented.

4) Line 106-107 ...flexibility with many vibrational modes, as classification illustrates (Supplementary Figure 5) - but there are not many classes on Supplementary Figure 5, just two. Besides, 3D classification is omitted in the Supplement figures. It should be present.

We thank the reviewer for this comment. Indeed, we showed two classes at the extreme ends of the conformational analysis, where the difference is highlighted best.

We have now extended the manuscript with the graphs of the complete vibrational analysis, where the positions of each individual molecule can be seen (**Supplementary Figure 5**). As described in the manuscript, there are not discrete states, rather a fluent transition from the one state to the other.

This transition is now described in the cartoon **Supplementary Figure 5**.

5) Figure 3d does not differ from Supplementary Figure 5 in color, shape, and orientation. Assuming that in Supplementary Figure 5 a filled P116 is presented, how can we understand that Figure 3d reflects the refilled P116?

We thank the reviewer for pointing out the duplication. We have improved supplementary Figure 5 and associated it with the vibrational mode analysis. In addition, we have labelled Figure 3d, to make clear that this reflects the refilled P116.

6) Supplemental Figure 2b – why does the corrected curve on the FSC plot have such a ‘spike’ in the middle (purple)? Could it be a result of overfitting?

We thank the reviewer for pointing this out. The spike in the corrected FSC occurred through the combination of the low frequencies FSC and the bias-free high-frequencies. The corrected FSC used here, as published in Chen et al (High-resolution noise substitution to measure overfitting and validate resolution in 3D structure determination by SP cryoEM, 2013) aims at estimating the overfitting and providing an overfitting-free estimate of the resolution. Thus, overfitting should not be an issue. We have now used a looser mask for calculating the corrected FSC, and thereby the curve has no spike, and the resolution remains the same.

7) Supplemental Figure 2c – there is no explanation for why there are two maps present (different threshold?) and why the dimers are not symmetrical. Does it reflect C1 symmetry?

Indeed, the two maps are shown at different thresholds, which is now noted in the figure legend. We also updated the figure legend noting that this is C1 symmetry.

Supplementary Figure 2c now reads:

“Local resolution map at different thresholds of the cryoEM density map (C1 symmetry) of P116 ranging from 2.5 to 5 Å”

8) Line 192 and below: “P116 is conformationally flexible” - it has already been mentioned several times before that it is flexible, why repeat this again? The whole paragraph can be easily omitted.

We agree with the reviewer that this may be redundant, but also concisely summarized the flexible components, which we now also describe in the cartoon in **Figure 4f**.

We have removed redundancies and refined the paragraph.

The text now reads:

“Conformational flexibility of P116

The structures of the original P116, empty P116 and refilled P116 samples appear predominantly as homodimers. In all cases, the homodimer exhibits significant flexibility. Most prominently, the arc diameter of the empty structure is approximately 10 nm smaller to that of the

original and refilled structures. In addition, a wringing motion is visible in the refilled structure: each monomer is twisted in the opposite direction along the axis perpendicular to the dimer axis by ~80 degrees, and bends up to 20 degrees, depending on its cargo. (Figure 3d, Supplementary Movie 6 and Supplementary Figure 5). In all P116 structures, the N-terminal domain is the most flexible. Within the core domain, temperature factors are higher at the fingertips, indicating movement of the antiparallel α -helices. When the fingers approach the palm, this results in the core domain constricting and a clash with the densities therein (Supplementary Movie 12). “

9) Line 250 “Of ~46,000 particles that were identified as HDL, ~25,000 were attached to P116” – this is also not visible on Figure 4f. Classes are too small. Consider adding arrows to point to the extra-density, which belongs to HDL. Also, the gallery of free HDL particles is necessary.

We have updated **Figure 4** with the classes and the cartoon. Instead, we have removed the cartoon of [3H] experiment. We hope in this way to provide a clearer picture of the mechanism of lipid uptake.

10) Line 524-526 “Due to the flexibility of P116 and the variability of HDL, only one subunit of P116 can be seen at this threshold. Reducing the threshold causes the second subunit to appear.” – show this in the supplement.

We thank the reviewer for this statement. We show now the presence of the complete P116 in the classes gallery. Unfortunately, the isosurface presentation is not good enough for the illustration. Through the flexibility of the molecule the second subunit does not appear properly. For this we have removed the sentence from the manuscript to avoid misunderstandings.

11) On a tomogram in Supplementary Figure 10 only one curved density is visible; why it reflects an arrangement of P116 with respect to the Mycoplasma membrane is not clear. Since Mycoplasma is a tiny cell, it will be useful, instead of one tomogram, to collect more cryo-images of cell P116 proteins on the surface, and, again, demonstrate via a gallery.

We thank the reviewer for this suggestion. We now provide a gallery as suggested by the reviewer in order to demonstrate the arrangement of P116 (**Supplementary Figure 11**). Furthermore, we adapted the manuscript, to clarify, that we only expect 34 copies of the protein to be present in *M. pneumoniae*. The text now reads as following:

“Despite the essential role of P116, the M. pneumoniae genome contains only a single copy of p116 (mpn213) and on average only 34 copies of the protein are present in M. pneumoniae (14)”

12) In the Discussion section, Authors claim that they suggest a mechanism for extracting the lipids from the environment. Here, the schematic will be useful to illustrate their conclusions. We now provide a schematic of the mechanism in **Figure 4f**.

Reviewer #3:

1. It is not clear how the protein releases the lipids into the membrane of the bacteria and how does it identify the host from its own. Is it flexibly attached to the membrane of the bacteria ? How does it distinguish between the bacterial membrane and the host membrane ?

We thank the reviewer for their question, which is a kind suggestion for further studies. At this point, we do not yet know the mechanism of membrane discrimination. The only difference is that P116 is embedded in the bacterial membrane, which could facilitate the discrimination. We now provide an additional cartoon in Figure 4f and have rewritten the paragraph for a better understanding.

The text now reads:

“Cryo-electron tomograms of whole M. pneumoniae cells indicate that this region faces away from the M. pneumoniae membrane, thus being accessible to vesicles and lipids. This presents a possible explanation as to how P116 avoids extracting lipids from the M. pneumoniae membrane itself. However, the unambiguous identification of P116 on the M. pneumoniae membrane is challenging due to the low copy number of P116 (14), and further experiments are required to better characterize the attachment of P116 with the M. pneumoniae membrane (Supplementary Figure 11).”

2. Please clarify what do you meant by arbitrary view (Fig. 3b)

We improved the illustration and labelled it as a front/along the main axis view.

Decision Letter, first revision:

Message: Our ref: NSMB-A46822A

16th Nov 2022

Dear Dr. Frangakis,

Thank you for submitting your revised manuscript "Essential protein P116 extracts cholesterol and other indispensable lipids for Mycoplasmas" (NSMB-A46822A). It has now been seen by the original referees and their comments are below. The reviewers find that the paper has improved in revision, and therefore we'll be happy in principle to publish it in Nature Structural & Molecular Biology, pending minor revisions to comply with our editorial and formatting guidelines.

Kind regards,
Florian

Dr Florian Ullrich
Associate Editor, Nature
Consulting Editor, Nature Structural & Molecular Biology
ORCID 0000-0002-1153-2040

Reviewer #1 (Remarks to the Author):

The major issues I brought up in the previous review have been addressed.

Thank you for changing the title.

I have no more productive suggestions and I pass this to the editor.

Reviewer #2 (Remarks to the Author):

I am satisfied with the comments and changes done by the authors. Paper can be published in the current form.

Author rebuttal, second version

We would like to thank all reviewers for their very constructive comments and suggestions. No further changes were required.

Final Decision Letter:

Message 6th Jan 2023

:

Dear Dr. Frangakis,

We are now happy to accept your revised paper "Essential protein P116 extracts cholesterol and other indispensable lipids for Mycoplasmas" for publication as a Article in Nature Structural & Molecular Biology.

As soon as your article is published, you can generate your shareable link by entering the DOI of your article here: <http://authors.springernature.com/share>

. Corresponding authors will also receive an automated email with the shareable link

Your paper will be published online soon after we receive proof corrections and will appear in print in the next available issue. You can find out your date of online publication by contacting the production team shortly after sending your proof corrections. Content is published online weekly on Mondays and Thursdays, and the embargo is set at 16:00 London time (GMT)/11:00 am US Eastern time (EST) on the day of publication. Now is the time to inform your Public Relations or Press Office about your paper, as they might be interested in promoting its publication. This will allow them time to prepare an accurate and satisfactory press release. Include your manuscript tracking number (NSMB-A46822B) and our journal name, which they will need when they contact our press office.

About one week before your paper is published online, we shall be distributing a press release to news organizations worldwide, which may very well include details of your work. We are happy for your institution or funding agency to prepare its own press release, but it must mention the embargo date and Nature Structural & Molecular Biology. If you or your Press Office have any enquiries in the meantime, please contact press@nature.com.

Please note that Nature Structural & Molecular Biology is a Transformative Journal (TJ). Authors may publish their research with us through the traditional subscription access route or make their paper immediately open access through payment of an article-processing charge (APC). Authors will not be required to make a final decision about access to their article until it has been accepted. <https://www.springernature.com/gp/open-research/transformative-journals>> Find out more about Transformative Journals

Sincerely,

Katarzyna Ciazynska
(she/her)
Associate Editor
Nature Structural & Molecular Biology
<https://orcid.org/0000-0002-9899-2428>
